# Modular stimuli-responsive hydrogel sealants for early gastrointestinal leak detection and containment

Alexandre H. C. Anthis[1,2], Maria Paulene Abundo[3], Anna L. Neuer [1,2], Elena Tsolaki[1,2], Jachym Rosendorf [4,5], Thomas Rduch [2,6], Fabian H. L. Starsich[1,2], Bernhard Weisse [7], Vaclav Liska[4,5], Andrea A. Schlegel[8,9,10], Mikhail G. Shapiro [3,11] & Inge K. Herrmann [1,2] ✉

Millions of patients every year undergo gastrointestinal surgery. While often lifesaving, sutured and stapled reconnections leak in around 10% of cases. Currently, surgeons rely on the monitoring of surrogate markers and clinical symptoms, which often lack sensitivity and specificity, hence only offering late-stage detection of fully developed leaks. Here, we present a holistic solution in the form of a modular, intelligent suture support sealant patch capable of containing and detecting leaks early. The pH and/or enzyme-responsive triggerable sensing elements can be read out by point-of-need ultrasound imaging. We demonstrate reliable detection of the breaching of sutures, in as little as 3 hours in intestinal leak scenarios and 15 minutes in gastric leak conditions. This technology paves the way for next-generation suture support materials that seal and offer disambiguation in cases of anastomotic leaks based on point-of-need monitoring, without reliance on complex electronics or bulky (bio)electronic implantables.

Every year approximately 14 million people undergo abdominal surgery worldwide. These operations consist of the resection of diseased or unwanted tissue and the reconnection of the healthy remainders with sutures or staples. While these procedures are lifesaving for a multitude of diseases, ranging from tumor resections to weight loss inducing gastric bypass, they also carry significant risks. These threats are associated with the leaking of digestive or microbially active fluids through sutured or stapled anastomotic reconnections, days after a successful surgery and can vastly delay recovery[1]. Such anastomotic leakages are one of the most dreaded complications following abdominal surgery with reported incidence rates ranging from 4 to 21%[2], depending on the patient's condition and the surgeon's experience[3]. Reported mortality rates for patients suffering from anastomotic leaks range from 6 to 27%[4]. In the much-feared progression to septic peritonitis caused by the leaking of the bacteria-containing intestinal fluid, mortality rates as high as 50% have been

[1]Nanoparticle Systems Engineering Laboratory, Department of Mechanical and Process Engineering, ETH Zurich, Sonneggstrasse 3, CH-8092 Zurich, Switzerland. [2]Laboratory for Particles Biology Interactions, Department Materials Meet Life, Swiss Federal Laboratories for Materials Science and Technology (Empa), Lerchenfeldstrasse 5, CH-9014 St. Gallen, Switzerland. [3]Division of Chemistry and Chemical Engineering, California Institute of Technology, Pasadena, CA 91125, USA. [4]Department of Surgery, Faculty of Medicine in Pilsen, Charles University, Prague, Czech Republic. [5]Biomedical Center, Faculty of Medicine in Pilsen, Charles University, Prague, Czech Republic. [6]Department of Gynaecology, Cantonal Hospital St Gallen (KSSG), Rorschacherstrasse 95, CH-9007 St Gallen, Switzerland. [7]Laboratory for Mechanical Systems Engineering, Department of Engineering Sciences, Empa - Swiss Laboratories for Materials Science and Technology, Ueberlandstrasse 129, CH-8600 Dübendorf, Switzerland. [8]Department of Visceral Surgery and Transplantation, University Hospital Zurich, CH-8091 Zurich, Switzerland. [9]Swiss HPB and Transplant Center, Zurich, Rämistrasse 100, CH-8091 Zurich, Switzerland. [10]Fondazione IRCCS Ca' Granda, Ospedale Maggiore Policlinico, Centre of Preclinical Research, Milan 20122, Italy. [11]Howard Hughes Medical Institute, Pasadena, CA 91125, USA. ✉e-mail: ingeh@ethz.ch

reported[5]. Such complications make the treatment of gastrointestinal anastomotic leaks especially costly (+30k USD per case), lengthy (+10 days in hospital) and complex[6].

To determine or infer leaking of sutured reconnections, surgeons rely on a vast array of criteria and equipment with relatively poor sensitivity and specificity. Such criteria include the assessment of manifested clinical symptoms such as tachycardia, hyperthermia, and oliguria once developed to a detectable extent. Additionally, they include the measurement of surrogate markers, such as C-reactive protein monitoring[7] and can extend to the assessment of the mental status of patients and more[8]. All the aforementioned criteria are highly indicative only after leak induced deteriorations have taken hold and the condition has become symptomatic. Thus, despite best efforts, the early (before severe clinical symptoms arise) and unambiguous identification of a gastrointestinal anastomotic leak remains challenging, while also presents itself as a seminal bottleneck in the fight against leak-related mortality[9,10]. To impart greater safety during rehabilitation and enhance patient monitoring methods, the installation of semi-permanent drains[11] following surgery and for the duration of recovery is commonplace. Additionally, the diversion of tissues containing digestive content to the exterior surface of the abdomen (temporary stomas)[12], are widely used measures in the struggle against digestive leaks. These measures, thus, underline both the gravity of potential complications from an anastomotic leak, as well as the inexistence of adequate technologies for early detection of deteriorations at the level of the operation site and patient.

Material innovations at the level of staplers and suture supports aim in the first place at preventing the site from leaking as well as supporting the healing of the sutured tissue. However, despite several surgical adhesives and sealants on the market, clinical success of these products is vastly limited in the case of gastrointestinal anastomotic leaks[13]. Tachosil[14], a widely used suture support in reconnections within the abdominal area, presents little benefits in the case of abdominal anastomotic leaks as fibrin-based materials are prone to digestion[15]. Numerous adhesive technologies and patches were recently developed with the prospect of sealing tissues in various places of the body. Such technologies have been designed leveraging N-hydroxy succinimide interface activation[16,17], dopamine coordination[18], layering and folding[19] and more[20,21]. While key metrics of tissue sealant performance such as tissue adhesion and bio-compatibility are central focal points for researchers[22], the performance of these materials is only rarely assessed while in contact with active and digestive effluents (including intestinal content containing bacteria, intestinal fluid and bile). Additionally, contemporary tissue adhesives only have limited ability to support the healing[23] while at the same time completely lack appropriate monitoring capabilities. First attempts into suture monitoring (absent of sealant properties) have most recently been undertaken, however, these fail to offer leak containment and rely on implanted electronic pledgets and bulky radio-frequency setups, which require significant engineering before clinical translation can even be considered[24].

The present work reports on the design and application of the first-of-its-kind gastrointestinal leak sensing hydrogel sealant, capable of incorporating both therapeutic and monitoring elements within its structure thanks to its modular design. The biocompatible and layered patch adhesive, grafts on tissue in a tissue-compatible way based on a mutually interpenetrating network (mIPN) traversing the hydrogel patch and the tissue simultaneously, ensuring leak containment and in-patch ultrasound contrast changes for swift detection of impending leaks. The employed hydrogel networks are thus selected not only to assure residence of the sealant in place under most demanding conditions but also to enable monitoring capabilities of sutured reconnections and for the early recognition leaks directly on various tissues of interest of the gastrointestinal tract. Sensing of the described leaks is made possible using two distinct types of sensing elements, namely a pH-responsive agar matrix and/or the incorporation of enzyme-responsive echogenic sensing entities into a soft polyacrylamide matrix. The deployment of sensing elements activating or de-activating in an enzymatic or pH triggered manner, respectively, showcases the potential for disambiguation in the detection of anastomotic leaks, early in their development trajectory. Taken together, we herein introduce smart surgical sealants that remain firmly attached to the tissue even under harsh digestive conditions, and feature integrated leak-sensing capabilities, enabling inexpensive, point-of-need monitoring, accessible using a smartphone-operated ultrasound probe, allowing prognosis of impending digestive leaks.

## Results

### Design of the gastrointestinal anastomotic leak patches with integrated sensing elements

The envisaged application of the leak-detecting hydrogel sealant patch, during a resection and anastomosis surgery, enabling the non-invasive post-operative monitoring of the anastomotic site, is illustrated in Fig. 1a. Firm and durable tissue adhesion of the hydrogel sealant under conditions of leak is imparted to the system by the use of a mutually interpenetrating network (mIPN). This firm adhesion under digestive conditions enables the detection of leaks before these even develop. The mIPN immobilized sealant contains carefully selected sensing elements in order to offer unique monitoring capabilities and detection of leaks using easily accessible ultrasound imaging. The molecular make-up, geometry and application of the layered hydrogel patches is displayed in Fig. 1b. The hydrogel patches are prepared using free radical polymerization, layer upon layer and are highly modular in character and architecture. Layers and matrix properties were chosen in order to allow both optimal functioning of the sensing elements and assure optimal sealing, simultaneously. More specifically, the patches are composed of an outermost non-adhesive backing. This latter is fused with an adhesive layer which houses sensing elements (in a pattern of choice) and may also comprise therapeutic elements contained within the same layer (Fig. 1c). The non-adhesive backing is made of a 50 wt% poly(N-hydroxylethyl acrylamide) (PNHEA) hydrogel for the avoidance of post-surgical adhesions[25]. The adhesive part, consisting of superabsorbent polyanionic polymer poly(2-acrylamido-2-methyl-1-propanesulfonic acid) sodium salt (PAMPS)[26] at 50 wt% forms the tissue contact layer. Thanks to PAMPS this latter layer is highly extensible (vide infra) and provides an initial adhesive contact with the tissue thanks to its highly charged character. Within the PAMPS contact layer, localized sensing elements capable of scattering ultrasound either due to their intrinsic nature as hydrogel-dispersed sensing materials (i.e: gas vesicles) or upon creation of scattering gas bubbles following contact with gastrointestinal fluid, were explored. We present herein, two types of conceptually distinct sensing elements: (i) enzymatically digestible gas-filled proteinaceous structures (gas vesicles derived from Halobacterium salinarum (termed Halo GVs)[27], for TurnOFF sensing) and (ii) matrix dissolved acid-reactive sodium bicarbonate (for TurnON sensing). These materials are respectively embedded either in a soft acrylamide matrix in the case of gas vesicles, or in a 2 wt% agar matrix. In the first case polyacrylamide allows the biologically derived gas vesicles to avoid instant degradation due to matrix conditions (in sharp contrast to other matrices) while at the same time allows for the maintenance of distinguishable ultrasound signal in conditions absent of digestive enzymes. In the second case, agar, a highly porous macromolecular network, allows the accumulation of $CO_2$ gas bubbles, after contact with gastric pH effluents and within the patch's structure, resulting in an efficient and readily observable ultrasound signal, all while maintaining integrity under normal physiological conditions. Taken together, the modularity of the layered synthesis even allows for the optional incorporation of therapeutic elements such as antimicrobial materials or compounds, properties imparted to the system following

the illustrative embedding of ZnO[28] nanoparticles or gentamycin in a 20 wt% polyacrylamide hydrogel.

The prepared double-layer, sensing element loaded, sealant patches are then attached to target tissues using a mutually interpenetrating network technology (mIPN), which firmly anchors the sealant patch on the tissue and enables long lasting adhesion and leak avoidance under harshest digestive conditions. Here, the mutually interpenetrating network precursor consists of a solution of 33 wt% N-acryloyl glycinamide (NAGA) and a visible light initiator Lithium phenyl-2,4,6-trimethylbenzoylphosphinate (LAP), in which the premade layered hydrogel (containing the sensing elements) is soaked prior to the attachment to tissue.

Upon patch application to the tissue of interest, the constituents in the anchoring monomer solution form a mutually interpenetrating network (mIPN) after light irradiation. The formed network transverses both the hydrogel patch and the tissue, joining them together. In contrast to our recent earlier work[15], where mIPN composition was tailored to maximize chemical interactions with the tissue, the herein explored mIPN system showcases the use of PNAGA, a widely explored non-adhesive polymer[29–31] in the role of a uniquely stable and robust tissue adhesion network. Additionally its use allows the anchoring to occur in absence of crosslinkers[31] and potentially harmful UV light[32,33] all while maintaining integrity and functioning of sensing elements and

modalities. As such, the attachment of the prepared hydrogel patches via a mIPN assures the long-term high-performance grafting of the sealant to tissues (>24 h direct contact with simulated intestinal fluid (SIF) or simulated gastric fluid (SGF), see Fig. S1). These properties overall being achieved by harnessing the synergistic effect[15] between the intrinsic properties of each functional layer (swelling, porosity etc) and those of the tissue interface, all together amounting to chemical interactions (electrostatic, hydrophobic etc.) and mechanical fixation (vide infra). The tissue anchoring strategy based on a PNAGA mIPN allows, as a result, the versatile application of the patch to various tissues of the abdominal cavity such as the stomach, small intestine and colon, without compromising sensory or adhesive performance under digestive conditions. All together amounting to a material specifically engineered for leak protection and detection, for tissues of the entire abdominal cavity (Fig. 1d).

### Tissue adhesion to abdominal cavity tissues via PNAGA mIPN

After application to tissues, adhesion energies of the attached patches are measured using a T-peel setup (Fig. 2a), energies of comparable range to first generation mIPN adhesives ($124 \pm 21$ vs. $92 \pm 24$ J/m$^2$)[15] are observed, indicating a formulation with comparable performance but substantially improved biocompatibility for intestinal tissue. Interestingly, the second generation layered hydrogel system not only yields

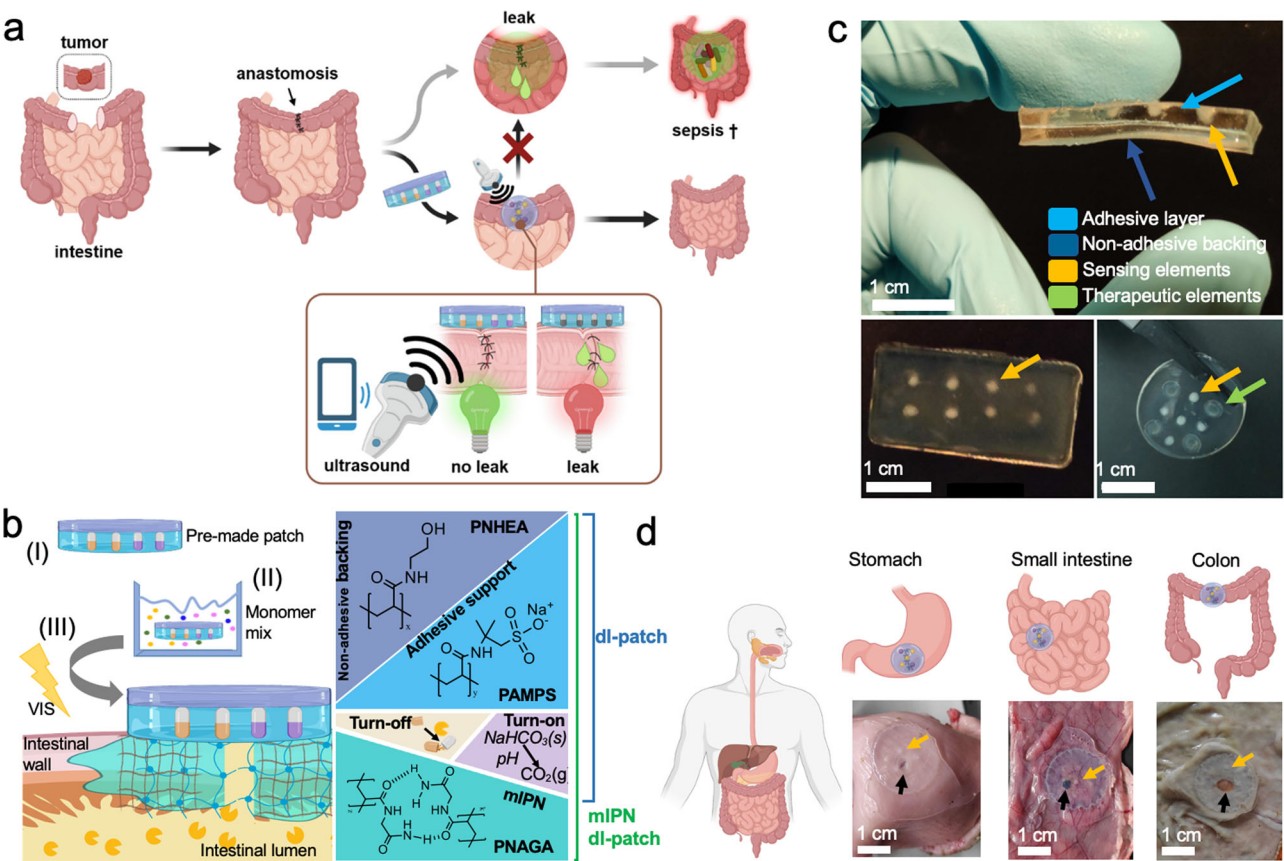

**Fig. 1 | Smart sealant patch design. a** Envisaged application of the leak-detecting sealant hydrogel patches enabling non-invasive postoperative surveillance of anastomoses using port-able, pocket handheld ultrasound probes (cross denotes high mortality of developed sepsis). **b** Composition of the layered-leak detecting sealant hydrogel patch, composed of a non-adhesive backing (PNHEA), adhesive suture support layer (PAMPS), sensing elements (TurnOFF sens-ing elements comprising enzyme-digestible gas vesicles or TurnON elements comprising pH-reactive sodium bicarbonate) attached to tissue via the formation of a mutually interpenetrating network (mIPN). **b-I** Application of the as-prepared hydrogel patch, which is incubated in a 33 wt% NAGA monomer/water solution containing LAP initiator (b-II) and finally is attached to tissue using visible light to form an

mIPN (lightning denotes light) (b-III). **c** As-prepared hy-drogels, manufactured in various shapes, exhibit discrete layers and compartments, namely adhesive sup-port (light blue arrow), non-adhesive backing (dark blue arrow), TurnOFF Halo gas vesicles sensing elements (5 µL 20 vol%, orange arrow) and 2.5 mg/mL ZnO in PAAm an-tibacterial elements (green arrow). **d** Illustration showcasing gastro-intestinal organs and the application of the hydrogel patches containing sensing elements (orange arrow) on the outer lining of porcine stomach, small intestine and colon, each closing a 4 mm defect (black arrow). Figure wide color coding: light blue—adhesive support, dark blue—non-adhesive backing, yellow—Turn-OFF sensing element, purple—TurnON sensing element, turquoise green—mIPN. Figure 1a, b, d has been created using biorender.com.

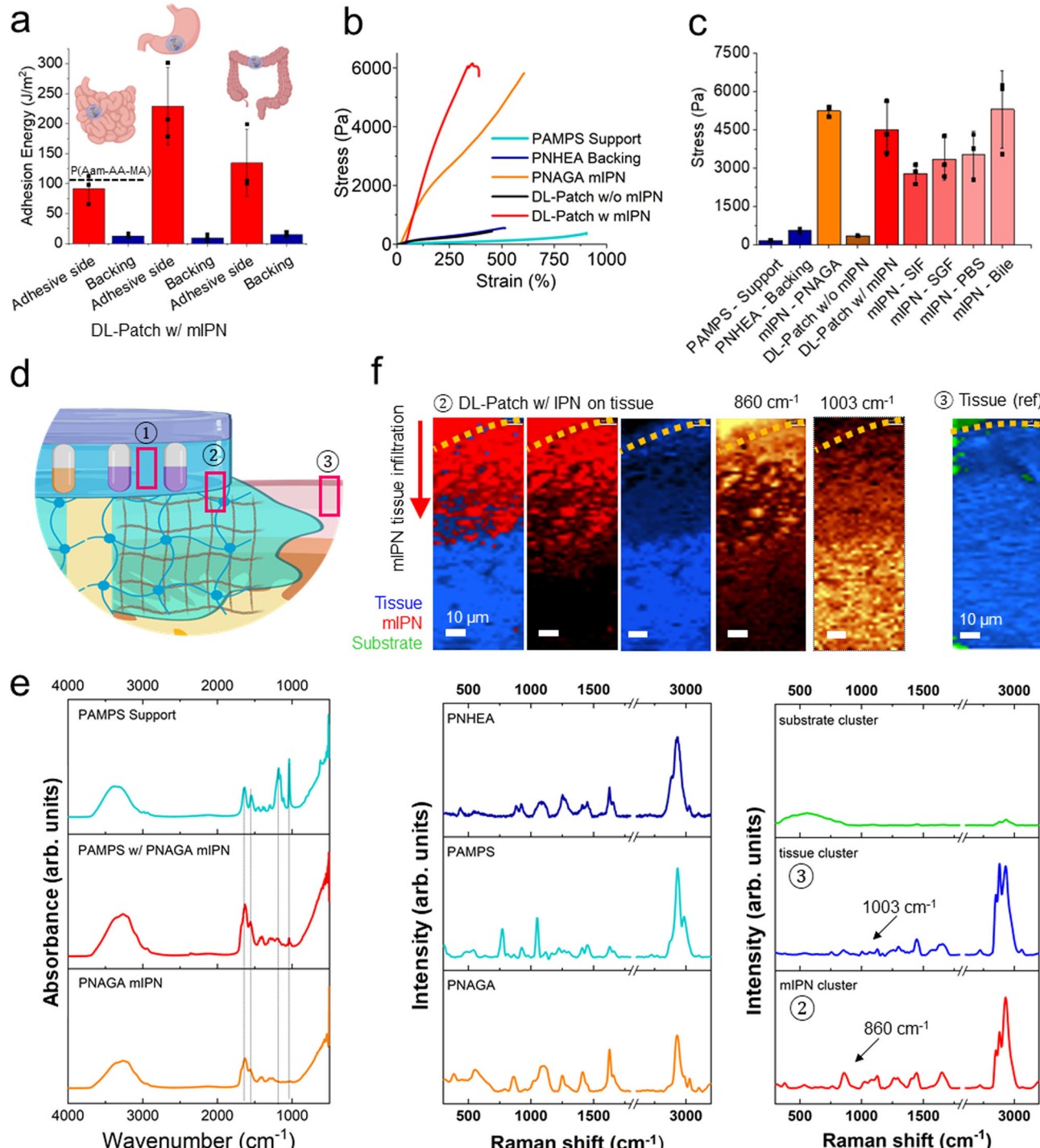

**Fig. 2 | Characterization of tissue adhesion and mIPN. a** Adhesion energy in the T-peel setup of the double-layered hydrogel patches after application to porcine small intestine, stom-ach and colon and corresponding PNHEA backing adhesion energies. $n = 3$ (independent ex-periments). Data shown as mean ± standard deviation. **b** Representative tensile strength curves of individual and fully assem-bled components of the hydrogel sealant. **c** Tensile stress of different components of the layered hydrogel system and stress modulation of PNAGA mIPN following incubation in various biological fluids. $n = 3$ (independent experiments). Data shown as mean ± standard deviation. **d** Assessment of the mutually interpenetrat-ing network charac-ter by vibrational spectroscopy. Sampling locations are indi-cated in red rectangles. **e** FTIR measurements (location ①) indicating PNAGA traversing in the PAMPS layer by comparison of PAMPS support as prepared and PAMPS support side after mIPN attachment. Dotted lines indicate 1039 cm-1 (S-O),

1183 cm-1(PAMPS characteristic peak), 1557 cm-1, 1633 cm-1 char-acteristic amide peaks of PNAGA35. **f** Raman spectromicroscopy (location ②) and corre-sponding k-means clustering maps and spectra of histological sections of patch sealed small porcine intestine. Peak maps of the phenylalanine peak (1003 cm⁻¹) unique to tissue and a PNAGA peak (860 cm⁻¹) unique to the mIPN are displayed along with cluster 1 (biological tis-sue, in blue), cluster 2 PNAGA mIPN (polymer-rich, in red), and the overlay. Map of native in-testine (location ③, overlayed tissue cluster in blue, sub-strate cluster in green). Corresponding Raman spectra of the different clusters and pure materials (PNAGA mIPN, PAMPS support and PHEA backing) for reference. Figure wide color coding: intense red−fully applied patch using PNAGA mIPN, orange−PNAGA mIPN, red shading−PNAGA mIPN under different biological fluids, dark blue−non-adhesive backing, light blue−PAMPS adhesive support. Figure 2a, d has been created using biorender.com.

even higher adhesion energies at the level of the colon and stomach tissues (134 J/m² and 229 J/m² respectively), but also exhibits minimal adhesive properties from the as intended non-adhesive side (backing). This illustrates the suitability and potential of the sealant technology at addressing leak prone reconnections at the level of multiple tissues of the abdominal cavity all while indicating minimal adjacent tissue interactions of the backing.

The tensile investigation of the developed layered hydrogel system further shines light on the role and synergy of each of the components of the layered sealant (Fig. 2b). While the PAMPS support layer is highly extensible and soft (strain: $967 \pm 56\%$, stress: $169 \pm 12$ Pa), its combination with a PNHEA backing restricts the resulting pre-application patches extensibility while yielding a reinforced double sided patch (strain: $551 \pm 78\%$, stress: $554 \pm 71$ Pa). Following application and attachment of the patches using a PNAGA mIPN yields a drastic enhancement in tensile strength, comparable to that of the PNAGA mIPN alone ($4515 \pm 1033$ Pa vs. $5245 \pm 210$ Pa). In this way, a material that is multifunctional, mechanically compatible and stably adhered to tissue is created. Interestingly, the maintenance of high tensile strength is largely retained even after 24 h incubation in various digestive fluids (see Fig. 2c). This further supports the pivotal role of the chosen mIPN as a robust anchorage within the superficial tissue layers (serosa, see also Fig. S2), especially when in direct contact with strongly digestive effluents and under conditions where all other available tissue adhesion strategies, including the ones based on mucoadhesion, fail.

Furthermore, to demonstrate the presence and mutually inter-penetrating network character of PNAGA inside the hydrogel patch following application, Fourier transform infrared spectroscopy (FTIR) and Raman spectromicroscopy were employed (Fig. 2d–f). Resultingly, the traversing character of the connecting network is elucidated in its entirety. On the one hand, the interpenetration of PNAGA, with the PAMPS support layer network is made evident by a decrease in intensity of peaks at the 1039 cm⁻¹ and 1183 cm⁻¹ level (corresponding respectively to the sulfonic groups and characteristic peak[34,35]) but also thanks to a pronounced increase in the corresponding amide functionality peaks (Fig. 2e).

After confirming the interpenetration of PNAGA networks into the premade hydrogel patch, Raman spectromicroscopy not only pro-vided clear proof for the extension of PNAGA network into the tissue, it also allowed quantification of the extent of the penetration into the serosa of different tissues (stomach [$64 \pm 4$ μm] >colon [$49 \pm 10$ μm] > small intestine [$35 \pm 4$ μm]). Expectedly, PAMPS and PNHEA were not found within the tissue despite making up the adhesive and backing layer of the sealant patch, in this way confirming the nature of the anchoring network and the discrete character of each layer. Thus, these two observations (based on Raman and FTIR) provide evidence in support of the interlocking and bridging character of the mIPNs, proving the mutual character of the interpenetration, and underlining a fine correlation between network-tissue penetration depth and pre-viously described adhesion energies.

## Patch layer synergy and properties upon exposure to gastrointestinal fluids

Further investigation into a sealant patch cross-section using scanning electron microscopy (SEM), shines light on the defectless joining of the various layers and different compartments into a single material (Fig. 3a). More specifically, in backscattering mode, one can distin-guish the different contrasts between the backing and the adhesive support imparted by the presence of sodium salt in the adhesion layer. Moreover, the location of the ZnO containing element is easily iden-tifiable thanks to its hemispherical shape, high z-contrast and a com-position which was confirmed by energy-dispersive x-ray spectroscopy (EDXS, see Fig. S3). Importantly, all interfaces indicate a smooth interlocking and clear compartmentalization resulting from the layer

by layer synthesis and as such presents no visible voids, cracks or otherwise weak points at the level of interpenetration between the backing and adhesion layer or between the adhesion layer and the ZnO compartment.

Further investigation into the collaborative behavior of those layers is performed by employing a 1 g patch (diameter 2 cm) to protect a 4 mm hole from the leakage of 7 g of enzyme rich simulated intestinal fluid (see Fig. S1). This setup, consisting of a patch attached to a model tissue allows the observation of the material's behavior when exposed to relevant biological fluids under most-demanding (open hole) and well standardized conditions. It showcases the dynamic behavior of the PAMPS layer (Fig. 3b). This latter absorbs intestinal fluid that makes its way to the patch and gives rise to a highly porous (suture) support contact layer, while the backing retains a smooth and non-altered structure, acting as a non-adhesive sealant.

The swelling properties of the PAMPS support layer, PNHEA backing, PNAGA mIPN as well as the whole patch as applied, were further studied in the presence of simulated intestinal fluid (Fig. 3c) as well as that of simulated gastric fluid, bile and phosphate buffer saline (PBS) (see Fig. S4) as a function of time. From those it can be observed that PAMPS, as expected, acts as a superabsorbent network[36], when alone. When traversed by a minimally swellable (in all biological fluids) PNAGA network, swelling overall is drastically reduced and thus brought to lower non-unitary levels, confirming the increase in stiff-ness observed after application of the patch due to the presence of the PNAGA mIPN (see Figure S5). This exhibited property underlines firstly the stability imparted to the overall network by the presence of PNAGA as well as the maintained ability of the suture support layer to absorb biological fluids and consequently allow digestive substances within the patch. The same properties also allow the introduction of ther-apeutics (such as antimicrobial[37,38] ZnO nanoparticles, see Fig. S6 for characterization data), while importantly, enable trigger-responsive ultrasound sensing elements that activate or de-activate based on the biological fluid in contact with the layered sealant. Interestingly, in the case of the optional addition of a therapeutic element, the use of a polyacrylamide matrix within the PAMPS suture support layer allows for the stimuli-responsive release of $Zn^{2+}$, a known antibacterial ion[39], depending on the biological fluid in contact with the patch, or, alter-natively, integration of antimicrobial small molecule drugs, such as gentamycin (see Fig. 3d).

## Enzyme and pH-responsive echogenic sensing elements

Importantly, the insertion of trigger-responsive ultrasound probes was envisioned to allow early detection of leaking bodily fluids. Towards that end, protein-based gas vesicles (GVs) were investigated because of their longlasting signal stability as well as their ability to alter their echogenic and optical signal following digestion. This latter resulted in a loss of signal intensity following decomposition by enzymes of the intestinal tract (e.g. pancreatin, see Fig. S7). Alternatively, physico-chemical properties of biological fluids such as as pH can be harnessed in order to react inorganic carbonates, leading to the formation of echogenic $CO_2$ bubbles if entrapped in a gel matrix. Both approaches result in a detectable change in ultrasound scattering properties. These sensing elements are termed "TurnOFF" or "TurnON", depend-ing on whether the exposure to the respective body fluid leads to a decrease or an increase in ultrasound contrast. Figure 3d demonstrates the working principle of an enzyme-responsive TurnOFF element, where as-prepared prokaryotic organism-derived gas vesicles (dervied from Halobacterium salinarum and termed Halo GVs) embedded in polyacrylamide hydrogels were incubated in either PBS or SIF, respectively. The as-prepared gas vesicle-containing hydrogels, while when incubated in physiological conditions mimicking peritoneal fluid, remain opaque white for prolonged times (several months), become transparent after their exposure to enzymatically active SIF,

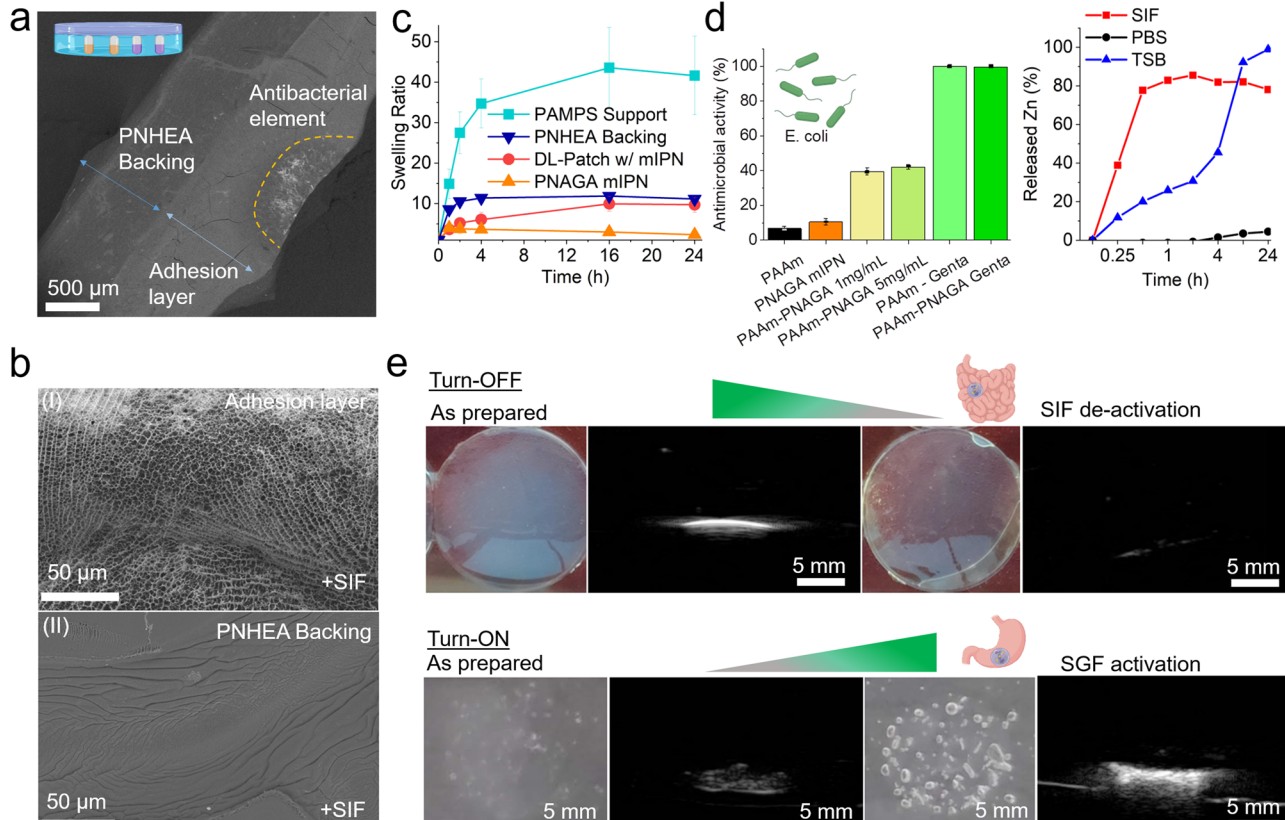

**Fig. 3 | Therapeutic elements and sensing. a** Scanning electron micrograph (SEM) of a cross-section of the as-prepared hydrogel patch generated in backscattering mode, showcasing the layered structure of the sealant composed of a backing, adhesion layer and antibacterial el-ement (ZnO (2.5 mg/mL) in PAAm). Samples were analysed in duplicate. Double edged arrows & dotted line denote beginning and end of corresponding layer. **b** SEM images of (I) porous character of the adhesion layer after contact with simulated intestinal fluid (SIF) during a 24 h sealing experiment, (II) non-porous character of the backing layer of the same sealant after 24 h. Samples were analysed in duplicate. **c** Swelling of individual layers as well as fully assem-bled hydrogel patch in simulated intestinal fluid at 37 °C (connecting lines for eye guidance). $n = 3$ (independently prepared hydrogel samples). Data shown as mean ± standard deviation. **d** Antibacterial activity against

E.coli and differential Zn release from double-layer gels as a function of biological fluid and time, TSB tryptic soy broth. $n = 3$ (independent samples). Data shown as mean ± standard deviation. **e** Functioning principle of the TurnOFF sensing element. Photographs and ultrasound images of 100 μL, 20 vol% Halo gas vesicle embedded in PAAm sensing element inside gel phantom; as-prepared and after deactivation following con-tact with SIF. **e** Functioning principle of the TurnON sensing element. Photographs and ul-trasound images of sodium bicarbonate in 100 μL agar gels embedded in a gel phantom as prepared and after contact with simulated gastric fluid (SGF). Figure wide color coding: light blue−adhesive sup-port, dark blue−non-adhesive backing, orange−PNAGA mIPN. Figure 3a, e has been created using biorender.com.

indicating digestion. Importantly the acoustic signal, generated by the introduction of these elements in an ultrasound phantom shows the transformation from bright and easily distinguishable (in presence of PBS), to dark and weakly scattering materials following exposure to simulated intestinal fluid. Moreover, further investigation into the enzymatic degradation of gas vesicles by pancreatin indicated the degradation of these latter in the span of 1.5 h at enzyme concentration approximating those of intestinal fluid, thus further confirming the enzymatic digestion as the main route of de-activation of the turning-off ultrasound monitoring elements (see Fig. S7).

TurnON sensing elements responsive to pH were also developed as alternatives to TurnOFF signals. They make use of a technology based on the combination of agar and dissolved sodium bicarbonate. Sodium bicarbonate thus, readily converts itself to $CO_2$ as soon as the liquid surroundings of the hydrogel experience a significant decrease in pH. The embedding of this latter in agar[40] ensures the entrapment of generated $CO_2$. As seen in Fig. 3e the as prepared elements scatter ultrasound only minimally in their native state, while upon incubation in gastric fluid gives rise to the quasi-immediate appearance of mac-roscopic bubbles within their structure. The ultrasound scattering intensity is greatly amplified, turning in this way the sensing element on.

## Cell and tissue compatibility of the leak-sensing sealant patch

With the sensing elements' working principle established, the layered formulation including its sensing elements and mIPN mediated attachment method was studied as per its interaction with different tissues and cells. In Fig. 4, the cytocompatibility and tissue interfacing characteristics of the applied patch are assessed using toxicity and viability assays in addition to histology of stomach, small intestine and colon porcine tissues. In order to characterize the cytotoxicity of the as-applied formulations, the lactate dehydrogenase (LDH) release of fibroblast cells as well as their viability was measured upon exposure to hydrogel-conditioned cell culture medium (Fig. 4a, b). Following an established evaluation method[41], as prepared and as-applied hydrogels were incubated with small quantities of cell medium simulating the gradual incorporation of the materials to the application area. The perfusion was iterated four times, and the cell culture medium col-lected was then added to the human primary fibroblast cells for lactate dehydrogenase (LDH) release and cell metabolic viability assessment. The data indicates negligible cytotoxicity from the first iteration onwards with as little as <6% toxicity and >75% metabolic activity compared to the respective control. These results indicate cyto-compatibility of the as-prepared materials from synthesis to application.

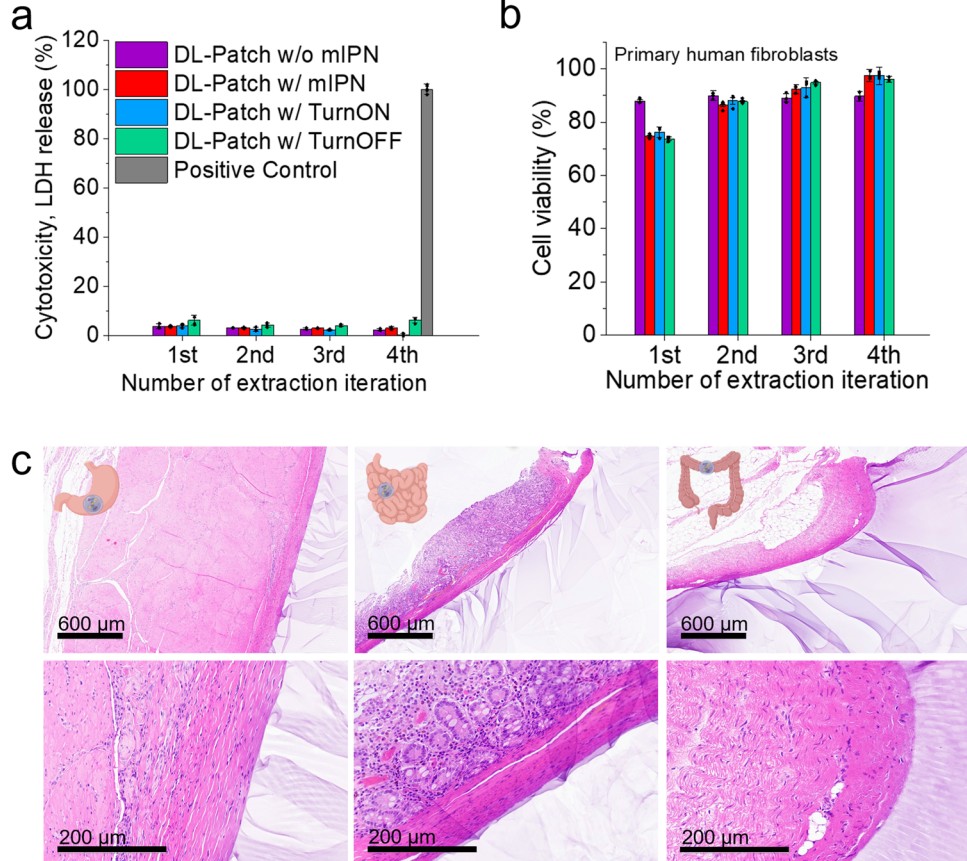

**Fig. 4 | Cell and tissue compatibility. a** Lactate dehydrogenase (LDH) release cytotoxicity assay of fibroblasts treated with conditioned cell medium as a function of perfusion iterations of the sealant patches with 5 mL of medium. Positive control (positive control (PC), 100% lysis, 0.1% TritonX) shown in gray. $n = 3$ (independent samples). Data shown as mean ± standard deviation. **b** Corresponding cell viability of fibroblasts based on metabolic activity using an ATP dependent Cell Titer Glo assay. Metabolic activity values expressed relative to untreated control. $n = 3$ (independent samples). Data shown as mean ± standard deviation. **c** Attach-ment and interfacing of hydrogel sealant on tissue as showcased by H&E stained biopsy sec-tions of porcine stomach, small intestine and colon at two different magnifi-cations. Histology samples were prepared in duplicate. Figure wide color coding: purple−double layer patch without mIPN, intense red−fully applied patch using PNAGA mIPN, blue−fully applied patch using PNAGA mIPN including a TurnON sensing element, turquoise green−fully applied patch using PNAGA mIPN including a TurnOFF sensing element, dark gray−positive control. Figure 4c has been created using biorender.com.

Furthermore, Fig. 4c shows the hematoxylin and eosin (H&E) staining of porcine stomach, small intestine and colon once sealed with sensing element equipped hydrogels. The firm interlocking of the hydrogel sealant with each of the tissues is made apparent by the firm dressing of the exterior lining of each tissue's serosa with the hydrogel. Upon closer examination, both freshly applied and after 2 h of contact with digestive fluids showcase stable and firm bonding to the exterior of the tissue wall, while leaving the inner structures intact and identical to the control (for bare tissue without sealant, see Fig. S8). Thus, these results provide a first indication for biocompatible high tissue adhe-sion in line with the tensile tests for three different tissue types using a mIPN mediated grafting.

## Tissue sealing and adhesion capabilities under most demanding conditions

For leak sensing to take effect and for the boundaries of the layered patches to be pushed towards a diagnostic and even prognostic class of sealants, successful and lasting leak sealing under digestive condi-tions and on tissues served as benchmark. The sealing capabilities of the multifunctional patches applied on porcine small intestine tissue were put to test under most demanding conditions by setting up an experiment, which brings in contact the patch with the tissue and its digestive fluid contents in a well-standardized manner. More specifi-cally, fresh porcine small intestine with a 4 mm hole (mimicking a

major anastomotic leak) is then sealed with a 2 cm wide circular layered patch grafted via mIPN and mounted on a two-compartment setup. The top compartment is then loaded with 7 g of biological fluids of interest, namely SIF, SGF and PBS. The system is incubated at 37 °C and orbitally shaken to simulate bowel movement. The leakage of these fluids is then monitored and compared to controls where tissues have no hole and no attached sealant. The sealant patches equipped with TurnOFF sensing elements and ZnO antibacterial compartments or TurnON sensing elements were found to perform equally or better than tissue without holes at mitigating leakage for up to 24 h of both simulated intestinal and gastric fluid (see Figure S1). More specifically, up to 8 h under the described conditions, no sample exhibits leakage in either of the categories defined by the incubating biological fluid. At the 24 h level, some leakage can be observed also in the corresponding controls, indicating increased permeability of the tissues over time. Despite the finite tissue lifetime, it is notable that all samples remained firmly attached to the tissue even after 24 h in direct contact with digestive fluids, underlining the high-performance adhesion and sea-lant capabilities of the developed patches. While these results are not fully representative of the even greater variety of gastrointestinal milieus present in different patients with different conditions, it is notable that these come in stark contrast to Tachosil sealed holes, which leak within minutes in the settings[15] explored and offer no diagnostic feature. Further light into the behavior of the sealant

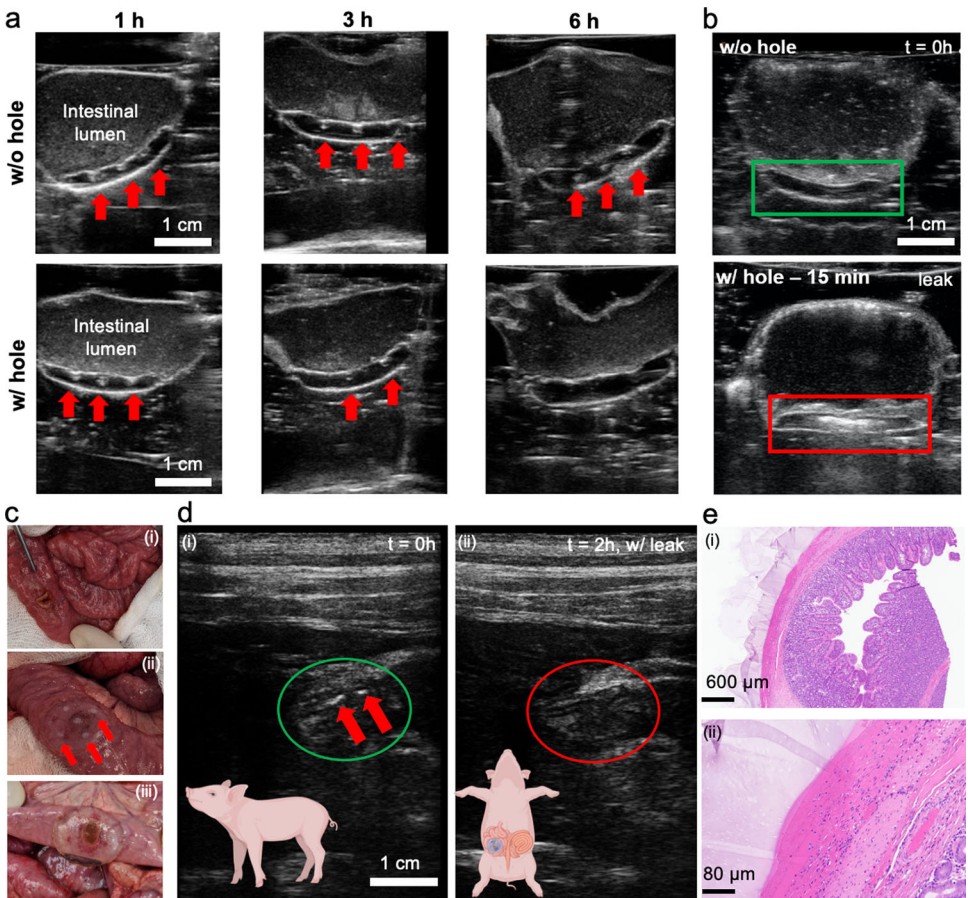

**Fig. 5 | Ultrasound-based leak detection ex vivo and in vivo.** Ex vivo ultrasound images obtained using an ipad controlled Clarius L7 HD probe of (**a**) a TurnOFF patterned hydrogel patch attached to intestine filled with SIF, equipped with a 4 mm hole simulating the perforation. The suture sites are monitored over six hours, showcasing pattern disappearance, N ≥ 3. **b** TurnON sensing element equipped hydrogel patch attached to an intestinal tissue model, filled with SGF indicating suture perforation by increased ultrasound scattering after fifteen minutes of contact with SGF, *n* ≥ 3. **c** In vivo application of TurnOFF DL-patches on piglet intestine with a defect. **c-i** 4 mm defect formation on live small porcine intestine. **c-ii** photograph of patch immediately after application on the defect. Sensing elements are clearly visible (red arrows). **c-iii** sealant patch 2 h post application on formed defect, presenting containment of digestive leak. **d** Ultrasound data recorded (**d-i**) immediately after patch application and surgical closure of the abdomen with a running suture. Sensing elements are discernible (red arrows). (ii), (**d-ii**) 2 h after application, sensing elements were not detectable anymore, in line with the disappearance of the opaque dots (**c-iii**). **e** Histological micrographs of intestinal tissue collected after euthanasia of the piglet shows firm attachment of the hydrogel to the serosa of the small intestine tissue and no visible tissue damage. Two samples were analysed yielding visually indistinguishable results. Figure wide color coding: red arrows—TurnOFF sensing element active. Figure 5d has been created using biorender.com.

patches under digestive conditions was shed using burst pressure and additional T-peeling experiments performed under digestive conditions (see Figure S1c, d). Adhesion under those conditions, including shaking and immersive digestive fluid contact was observed to retain at least 50% of its initial adhesion after 8 h of incubation under enzymatically active intestinal fluid contact, while in the case of gastric effluents (pH of 2.0), values remained close to levels upon application. Lastly, burst pressure was also investigated under conditions of digestive fluid contact, showcasing similar maximal burst pressures for both SGF and SIF conditions, overall greatly surpassing the native maximal intestinal burst pressure[42].

### Demonstration of leak sealing and detection capabilities in vitro and in vivo

With high sealant performance evaluated, the patch technology as a way to unambiguously identify an impending leak of digestive fluids was assessed. Firstly, the distinguishable character of the acoustic signal of TurnOFF sensing elements, compared to intestinal tissue, was put to the test and highlighted by the attachment of a gas vesicle loaded patch on a 1 m segment of porcine intestine (see Fig. S9). The intestine segment sealed by the patch was then placed into a box,

simulating the abdominal cavity with multiple layers of intestine on top of each other. Upon ultrasound examination, the patch is immediately identifiable and discernible from intestine alone, even by untrained eyes. Based on these results, each layered patch variation equipped respectively with TurnOFF or TurnON sensing elements was used to seal a tubular segment of small intestinal tissue containing digestive fluid (Fig. 5). The tissue was either left intact (control) or perforated with a 4 mm biopsy punch, this way aiming to simulate a breach of sutures. Enzyme-responsive TurnOFF sensing element loaded patches were exposed to SIF while pH-responsive TurnON sensing element loaded systems were exposed to SGF. The change in signal produced by the elements was monitored using a wireless, handheld Clarius 7 HD probe controlled by a portable device (smartphone or ipad). At discrete time intervals, the tissues were imaged by ultrasound imaging. The system's evolution was then followed under well-controlled ex vivo conditions in order to define both the functioning and the earliest moment an up and coming or occurring leak could be detected. From Fig. 5a, it can be observed that the samples bearing no hole retained their characteristic sensing element patterns over the entire timeline of the experiment. In contrast, the samples that experience a suture breach (and contain a 4 mm hole exposing the sealant to SIF)

gradually lose their characteristic pattern and sensing elements gradually disappear. This change in sensing elements pattern becomes already visible at the 3 h mark while it is even more evident at the 6 h time point, indicating that breaches of tissue by intestinal fluid could be detected in as little 3 h post breach. Importantly, the sealant patch remained firmly attached and effectively prevented leakage of digestive fluid. Furthermore, in the case of pH-responsive TurnON sensing elements monitoring SGF filled tissue for suture breaches (Fig. 5b), the activation of the sensing element appears even faster, with 15 minutes being enough to give a clear distinguishable signal in the case of patch-digestive fluid contact. Notably, the above experiments not only confirmed the optimal functioning combination of triggerable sensing elements and the sealant character of the patches via an mIPN, they also allowed for contrast changes to occur before any leak had taken hold, paving the way for sealants to become platforms of both diagnosis and prognosis potential. Taken all the above together, it is also important to observe some of the prevailing qualities of an mIPN designed for the purposes of sealing tissues under digestive leak conditions. These can be discretized as (i) being mechanically stable under harsh digestive conditions, (ii) having little to no interaction with the functioning of the patch or its components, (iii) having a composition that is not disruptive to tissue and the sealing interface.

Following confirmation of the patches being able to reliably report on the presence of digestive fluids ex vivo all while sealing tissues under demanding conditions, sensing patches were applied in an in vivo proof-of-concept setting (see Fig. 5c). More specifically, in a piglet model, a 4 mm defect was created at the level of the small intestine (Fig. 5c). The defect was then sealed with a TurnOFF circular layered hydrogel patch. Additionally, a reference patch was placed on intact intestine on the contralateral side and the abdominal cavity of the treated pig was closed by a running suture. The approximate application site was fiducially marked on the outer body of the hog. After closure, the patches were imaged and could be readily identified based on the characteristic ultrasound signal (Fig. 5d).

After identification, the model pig was allowed to rest under anesthesia for 2 h before re-assessing the state of the hydrogel sealants based on their echogenic signal. Upon examination, the signals of the defect bearing site were not discernible by ultrasound anymore, in contrast to the contralateral hydrogel patch on intact intestinal tissue. Subsequently, the abdominal cavity was opened and the locus of application was visually examined. The hydrogel sealants were found firmly attached to the intestinal tissue, effectively containing the leak (Fig. 5c, e and Fig. S10). Sensing elements in the patch sealing the defect were not easily discernible anymore by visual inspection of the patch, confirming the ultrasound observations and the feasibility of the sensing concept. Biopsy samples of native as well as layered hydrogel-sealed tissues were collected and histologically analysed. The resulting histological analyses indicate firm attachment of the hydrogel on the tissue serosa (see Fig. 5e) confirming previous in vitro results. No detectable tissue damage or alternations were found.

Overall, these experiments showcase the effective protection of the indicative sutured area, prompt sensing response of the layered adhesives upon gastrointestinal fluid contact and the straightforward distinguishability of the created signal compared to native abdominal tissue. In vivo investigations demonstrate the feasibility of the presented approach both from the sensing modality point of view as well as from the attachment method compatibility perspective. Taken together, the above results set the foundations for a technology that allows for the early detection of anastomosis deteriorations by physicians giving them the opportunity to intervene before digestive leaks occur, as well as adjust patient regimen based on patch integrity and signal. These latter also underscore and foreshadow the important role of sutures during events of deterioration, as in the inevitable case of a leak, they allow the limitation of leaked fluid volumes. Thus, while the development of technologies capable of offering sutureless surgeries, abdominal cavity operations shall be seen as in need for suture support rather than suture replacement.

## Discussion

This work presents a first-of-its-kind sealant suture support hydrogel patch with integrated leak-containment and monitoring capabilities for the early detection of leaks occurring at sutured reconnections within the abdominal cavity. The prepared layered and compartmentalized adhesives comprise a non-adhesive backing and an adhesive suture support layer containing triggerable sensing elements that TurnOFF or TurnON in a timely manner, depending on the digestive fluid that may be breaching the sutures. The layers of the as designed patch work in synergy to ensure easily discernible ultrasound signal changes all while remaining adhered on perforated tissues. They are attachable in a tissue-agnostic high-performance manner to various organ walls, ranging from the stomach to the colon. They do so by design, thanks to their grafting via a performance-optimized composition, mutually interpenetrating network. This network, made of PNAGA (employed here in a contradicting manner to its well-studied role as a non-adhesive barrier), is capable of bridging patch and tissues in a robust manner and in absence of crosslinkers. The combination of a polyanionic PAMPS contact layer and a PNHEA backing allows the former to develop a porous structure, despite being traversed by a robust PNAGA network, when in contact with biological fluids and thus make use of active substances within effluents, while leaving the backing intact. Thus, the contact with digestive fluid in synergy with the swellable adhesive layer of the patch, were used to turn-off ultrasound sensing elements composed of Halo gas vesicles via in-gel enzymatic digestion mediated by pancreatin. Alternatively, pH-responsive TurnON sensing elements made of an agar-bicarbonate formulation indicated contact with simulated gastric fluid. The sensing elements prepared being triggered via digestion or transformation in a matter of 3 h and 15 min for the TurnOFF and TurnON variations, respectively, pave the road to ultrasound monitoring of suture sites and sealing under conditions of leak through firm tissue anchoring. The robust attachment was also confirmed using biopsy cuts of fresh abdominal tissues and indicated firm and tissue-compatible attachment even in the presence of digestive fluids all while proving highly cytocompatible already from the first perfusion cycle on fibroblast cells. The designed patches were thus evaluated as per their sealing capabilities and indicated a high-performance leak containment, sustaining contact with SIF, SGF and PBS for up to 24 h with minimal leaking comparable or better than intact native tissue. The patches were further attached to tubular pieces of intestinal tissue, equipped with incisions simulating the severe breaching of sutures in postoperative scenarios, in an effort to assess the sensing modalities under the harshest controlled conditions. Ultrasound monitoring of these latter using a handheld probe and a tablet allowed for the identification of digestive intestinal fluid breaches as early as 3 h. While in the case in the case of gastric fluid and pH-responsive TurnON sensing elements signals of digestive effluent contact were clearly observable as early as 15 min post contact. Importantly, the signals produced by the patches were also easily discernible compared to native tissues. Proof of concept in vivo studies in a piglet model demonstrated the feasibility of the approach both from the sensing perspective as well as the anchoring method.

As such and while further animal studies and material optimization (fate and biodegradation) are envisaged, addressing both long term compatibility as well as performance optimization under digestive leak conditions, the herein presented technology paves the way for the modernization of suture supports. We presented materials that adhere in a high-performance, long-lasting manner, all while being capable of weathering contact with digestive fluids. Additionally, the combination of those sealants with sensory compartments within the patches offer entirely new capabilities and open the road to diagnostic

and prognostic sealants. These ones capable of informing on the state of the sutures and the necessity to intervene based on easy-to-read out and entirely biocompatible echogenic sensing moieties.

Additionally, the demonstrated and illustrative combination of sensing moieties with therapeutic elements such as antibacterials (ZnO, or gentamicin) gave rise to adhesives that could both detect but also potentially prevent infection in a stimuli responsive manner via the release of therapeutics. Continued investigations into synergies of those elements could give rise suture supports that release therapeutic payloads following an external trigger[43].

The development of intelligent materials such as the above presented can lead to the much-needed danger alleviation of peritonitis and sepsis, commonly caused by leaks of digestive fluid in the peritoneal cavity at the level of sutures. Additionally, with the signals of the developed patch being detectable with an inexpensive ultrasound probe and a tablet, the developed patch technology enables a radiation-free and potentially implantable drain-free routine monitoring of patients combined with the ability to unambiguously detect impending digestive leaks while on the move (ambulance) or at the hospital. All in all, the above layered suture support presents a versatile and holistic solution to suture leak monitoring applicable to a multitude tissues within the abdominal cavity.

## Methods

We confirm that our research complies with all relevant ethical regulations. Animal experiments have been approved by the Commission of the Medical Faculty of Pilsen, Charles University (project ID: MSMT-15629/2020-4), and are under control of the Ministry of Agriculture of the Czech Republic.

### Chemicals and materials

All materials were purchased from Sigma-Aldrich (Merck) with the exception of N-acryloyl glycinamide which was purchased from Abmole (Belgium). N-hydroxyl ethyl acrylate (NHEA) and (2-acrylamido-2-methyl-1-propanesulfonic acid) sodium salt (PAMPS) monomers were purified by passing them through a plug of basic alumina (Brockmann Grade I). Acrylamide, N,N′-Methylenebisacrylamide (mBAA) were used without further purification. Teflon molds for shaping hydrogels were manufactured in-house featuring various shapes and depths of 0.9 mm. Fresh porcine tissues were obtained from a local slaughterhouse (Schlachtbetrieb St Gallen, Switzerland). The intestine was cleaned of its contents manually and divided into pieces. Tissues were preferably used fresh or thawed once and subsequently used for experiments. Lyophilized ox-bile was purchased from Sigma-Aldrich and was reconstituted using milliQ water.

### Intestinal fluid preparation

Simulated intestinal fluid (SIF) was prepared using lyophilized pancreatin powder (>8 USP) and a protocol from the united states pharmacopoeia (Test Solutions, United States Pharmacopeia 30, NF 25, 2007) as previously used in other studies[44]. In brief, (6.8 g, 50 mmol) monobasic potassium phosphate was dissolved in 250 mL milliQ water. To this solution, (77 mL 0.2 mol/L) sodium hydroxide solution and 500 mL milliQ water were added and mixed along with (10 g) pancreatin (from porcine pancreas, 8 USP units activity/g). The SIF/P suspension was adjusted to pH 6.8 with either 0.2 mol/L sodium hydroxide or 0.2 mol/L hydrochloric acid and diluted with water to 1000 mL.

### Gastric fluid preparation

Simulated gastric fluid (SGF) was made using the guidelines from the United States pharmacopoeia. More specifically, a 35 mM NaCl solution was prepared from distilled water. The solution was then adjusted to pH = 2.0 using 0.1 M HCl and used straight away.

### Hydrogel preparation and assembly

In order to prepare the suture support hydrogel patches, stock solutions of each of the layerable components were prepared first and were sequentially added and polymerized into Teflon molds of desired dimensions. Towards the preparation of the sensing element layers, a polymerizable mix stock solution made of acrylamide monomer (AAm) was prepared by dissolving the monomer powder in milliQ water at 20 wt%. In parallel, a 2 wt% crosslinker solution was made by dissolution of (0.2 g, 1.3 mmol) N,N′-Methylenebisacrylamide (mBAA) dissolved in 9.8 g Milli-Q water. Both prepared solutions were kept for a maximum 30 days stored at 0–4 °C. Photoinitiator stock solutions (4.825 mg, 21 µmol) 2-Hydroxy-4′-(2-hydroxyethoxy)-2-methylpropiophenone (Irgacure 2959) were dissolved in 1 mL Milli-Q water, by 20 min sonication and were kept in the dark. By combination of the above stock solutions, the polymerizable basis and matrix for the sensing elements was prepared by mixing 5 mL of 20 wt% AAm, 108 µL of mBAA crosslinker stock solution and 500 µL of Irgacure 2959 (monomer/crosslinker ratio of 2000). From this resulting mix (20 wt% AAm, mBAA, Irgacure 2959), were prepared 20% vol gas vesicle (Ana or Halo) dispersions (1 mL gas vesicles: 4 mL 20 wt% AAm, mBAA, Irgacure 2959) which served as TurnOFF sensing elements.

Regarding antibacterial elements, the previously described AAm monomer polymerizable mix (20 wt% AAm, mBAA, Irgacure 2959) was used to prepare dispersions at the desired ZnO nanoparticle concentration (such as 2.5 mg/mL). Prior to polymerization dispersions were sonicated ensuring that particles were well dispersed prior to polymerization.

All polymerizable mixes accounting for sensing or antibacterial elements were then placed in the form of 5 µL and 12.5 µL volumes (Gas vesicle sensing element mix & ZnO antibacterial mix respectively) in a periodic arrangement of interest on Teflon molds (circular: d: 2 cm, thickness 0.2 cm & rectangular: 7 × 3 x 0.2 cm or 7 × 1.5 × 0.2 cm or 3 × 1.5 × 0.2 cm). The sensing element and antibacterial mix drops were then polymerized for 5 min using a UVASPOT 400/T mercury lamp at a distance of 30 cm from the source. The light source was also equipped with a filter (H2) allowing the spectrum interval from 300 nm till the visible to reach the polymerizable mix.

With the first discrete element layer prepared, an AMPS monomer mix consisting of 4 mL of 50 wt% of AMPS (aliquoted from the as received stock), 20 µL of mBAA stock and 30.6 µL of Irgacure stock solution, was prepared and used to add 300 µL of this latter on the polymerized sensing and antibacterial elements and the circular Teflon mold. After evenly spreading the mix and allowing the solution to settle for 1 min, the layer was polymerized as previously described for 5 min. This formed the suture support layer, typically in contact with the tissue.

With the suture support layer embedded with functional elements, in place, the same procedure was followed for the formation of the backing. Thus, using a 300 µL backing mix coming from a polymerizable stock solution composed of pure, inhibitor removed, 2 mL NHEA monomer, mixed with 2 mL of milliQ water, 53.32 µL of mBAA 2 wt% stock solution and 302 µL of the previously prepared Irgacure solution, the monomer mix was spread on the underlying support layer and left to diffuse in the support layers for 1 min before a 5 min polymerization step. The prepared patches were then kept in the mold and protected from drying using a polyethylene foil until application.

In a similar fashion but with a difference at the level of the incorporation of the TurnON sensing element ultrasound activable layered hydrogels were assembled. More specifically TurnON sensing elements were prepared by making a 2 wt% agar in water solution. This latter one, was heated to 100 °C and when the temperature was reached 2 wt% of NaHCO₃ was added to the stirring mix. After homogenization the solution was allowed to cool to room temperature and once gelled was cut in cylinders of 8 mm in diameter using an 8 mm sterile biopsy punch.

With the bulk of the sensing element prepared the cylindrical gels were cut into disks of 0.2 cm thickness using sterile scalpels. The prepared disks were then mounted in an 8 mm hole created at the level of a pre-polymerized PAMPS hydrogel layer to accommodate the sensing disk. The construct was fused together using 300 μL of the polymerizable NHEA backing mix as earlier described.

The mIPN polymerizable mix was prepared by mixing 2 g NAGA, 4 mL milliQ water, 450 μL 6.33 mg/mL LAP.

All prepared layered gels were foiled into the prepared molds and stored in at 0–4 °C after preparation and before use or used directly.

## Scanning electron microscopy

Hydrogel samples that were either freshly prepared or swollen in the cup model setting for 24 h in biological fluids were lyophilized for 24 h. The resulting pieces were mounted onto an SEM holder using carbon tape and coated with a (5 nm) layer of carbon using a Safematic CCU-010 coater. Scanning electron microscope (SEM) imaging was performed on a Quanta 650 SEM (FEI, Thermo Fisher Scientific) at an accelerating voltage of 10 kV.

## Swelling experiments

The relative swelling ratio $R_{rel}$ is defined as,

$$R_{rel} = \frac{M_s}{M_i} \qquad (1)$$

where $M_s$ is the mass of the hydrogel sample after swelling for a given time point, $M_i$ is the initial mass of the hydrogel sample. Initially 50 μL hydrogels were transferred into vials. Then (5 mL) of biological fluid ($N = 3$ for each fluid) were added respectively to investigate the dynamic hydrogel swelling in different physiological fluids. Finally, the falcon tubes were transferred onto a shaker (Titramax 101, Heidolph, 200 rpm) and incubated at 37 °C with shaking. The mass of the hydrogels was deduced by measuring the residual mass of the falcon after residual fluid removal. This was done at time points 1 h, 2 h, 4 h, 16 h and 24 h swelling. The average relative swelling ratios in different simulated body fluids were obtained at different time points.

## Zn²⁺ release in biological fluids and antimicrobial activity

ZnO containing hydrogels were placed inside 15 mL polypropylene falcon tubes. These were loaded with 7 mL of digestive or biological fluid of interest. After a given amount of time 0.5 mL of the incubating fluid was removed and replaced with fresh one. The extracted aliquots were further digested using a 2 mL 67% high purity $HNO_3$ and 1 mL of 30% high purity peroxide $H_2O_2$ in a microwave reactor. The resulting digest was diluted with milliQ water to 10 mL and was used to measure ICP-OES and determine the concentration of zinc for each time point. Measurements were carried out in triplicates. Antimicrobial activity against E. coli was measured using previously established protocols. Briefly, E. coli were exposed to extracts and bacterial growth was assessed by a kinetic measurement in a plate reader. Bacterial growth inhibition was expressed relatively to the gentamycin hydrogel control.

## mIPN formation−hydrogel application

All hydrogel samples prepared were applied on porcine tissue serosa. Hydrogels that were prepared as previously described were immersed for 10 min in 1.5 mL of a polymerizable mix water solution, consisting of 33 wt% NAGA monomer mix. For 3 mL of the desired mix 1 g of NAGA powder was mixed with 2 mL of MilliQ water and 225 μL of a 6.33 mg/mL LAP initiator solution. The incubation was performed in a 10 mL polyethyelene cup topped by a second plastic cup of the same size. This allowed to avoid drying and hydrogel curling due to swelling. After the incubation time was completed the tissue serosa was brought in contact with (150 μL for d = 2 cm circular patches) of polymerizable

fluid mix, at the area of application of the hydrogel (excluding the area of the (4 mm) hole, in the case of model experiments). The swollen hydrogels were then applied to the intestine making sure that no bubbles are trapped at the interface. Once in place a transparent glass plate was used to assure firm contact of the gel with the tissue surface and the samples were then subjected to 5 min of visible light irradiation using a 2x6W − 365 nm VL-206.BL lamp.

## Adhesion measurements

The adhesion strengths of the applied hydrogel patches on porcine tissues including stomach, small intestine and colon were investigated following the procedures described in the ASTM standard F 2256-05[45] (for the T peel test) and the ASTM standard F 2255-05[46] (for the lap-shear test). The tissues after being thawed at room temperature, were cut into pieces of 7 cm×1.5 cm using a scalpel. In order to limit the deformation of the gel and tissue[16], rigid transparent films (CG3720 color laser transparency film, 3 M) were used as backing layers, after being cut to the size of the applied hydrogel patches (5 cm × 1.5 cm × 0.2 cm) or intestine pieces and were attached to these latter with the use of a liquid superglue (Pattex Super Glue). The hydrogels were then applied to the intestine pieces as described earlier and the free ends of the hydrogel and the intestine piece were loaded to the instrument's clamps. Similarly, and to probe the adhesion energy of the backing post application, as described patches were applied in the indicated manner on mock Teflon setups. They were then flipped to bring the PNHEA backing in interaction with the tissue's serosa, which was sprayed with PBS to simulate 100% humidity of the abdominal cavity. A rigid backing was this time applied on the opposite site and the experiment was repeated as detailed. The loading rate was kept constant at 250 mm/min and the standard force-strain curves were recorded. The T peel strength was calculated as the maximum plateau value of the ratio between the standard force and width of the sample. At least three independent experiments were carried out per experimental group.

## Tensile measurements

Using custom dumbbell-shaped Teflon molds 4.5 cm in length and 0.5 cm in width tensile property tests of the various layers of the investigated patches were tested. Sample volumes were kept equal and layer volumes were adjusted to maintain comparable final sample sizes. The tensile properties were thus measured with a mechanical testing machine (Zwick/Roell Z100 (Zwick/Roell, Ulm, Germany)). All tests were done at a constant tensile speed of 5 mm/min. At least three independent experiments were carried out per experimental group.

## Rheological measurements

Dynamic shear oscillation measurements were used to characterize the viscoelastic properties of the layered hydrogel patches before and after application to tissue systems. These were done with as prepared patches but also after application to tissues and contact with digestive effluents at 37 °C. All the samples were fit to the rheometer (MCR 301, Anton Paar) receptacle size using a 25 mm Turnus Wad punch. Slippery samples due to swelling with biological fluids were immobilized using sandpaper at the level of the Peltier and rotating plate. All measurements were carried out at 37 °C with the oscillatory strain sweep measurements, performed at a constant frequency of 1 Hz while the oscillatory frequency sweep measurements were performed at a fixed constant strain of 1% over a frequency range of 0.01–10 Hz.

## Ultrasound phantom making for sensing element benchmarking

Phantoms were created using 20 wt% acrylamide polymerizable stocks. These comprised ammonium persulfate as the initiator, TEMED as the accelerant and were polymerized at 60 °C in an oven. A typical stock consisted of 500 mL 20 wt% AAm, 5.4 mL of mBAA 2%, 800 μL pure TEMED. The usual phantom making process consisted in the use

of a plastic rectangular mold 12 cm×6 cm x 6 cm to which an initial base layer of 60 mL of the above described mix was added. To that was added 2.240 mL of ammonium persulfate 60 mg/mL and the reacting system was then left to react for 5 min at 60 °C inside an oven. The base layer prepared, echogenic elements were placed in the order wanted on the surface of the first layer. To that was then added 210 mL of the phantom mix premixed with 7.84 mL of the ammonium persulfate initiator. After 5 min at 60 °C the phantom gel was removed from the mold and placed in a fridge at 2-4 °C. This allowed to remove air bubbles that appeared during mixing or degassing due to oven incubation. Measurements were performed the following day.

## DL-Patch−abdominal cavity tissue interaction biopsies
After obtaining freshly collected porcine intestine from Schlachtbetrieb St. Gallen, Switzerland, in situ, ex situ and Tachosil adhesives were applied on the exterior surface of the fresh porcine intestine (not perforated). After application the treated intestine was sampled using an (8 mm) biopsy punch. The collected biopsies were immediately placed in a 4% formalin solution. After 24 h in formalin, the samples were separated from the bulk of the swollen hydrogel adhesive, leaving the interface intact, using surgical scissors. The same procedure was followed in the context of a stationary, post-surgery simulating experiment, where the as prepared samples were brought in contact with SIF at 37 °C, 100% humidity and light agitation. Intestinal tissue was subjected to both pancreatin rich and depleted SIF as a control.

The collected biopsies were then sent to Sophistolab AG, Muttenz, Switzerland where they were block paraffinized, cut and stained using H&E. The samples were incubated for 5 min in Hematoxylin and 30 s in Eosin solutions. For preservation, the samples were dehydrated in steps (successively 70%, 80%, 90%, 96%, 100% EtOH, 2% isopropanol, 2 × xylene, 2 min each) and mounted for subsequent storage. A ZEISS Primovert Microscope with an Axiocam 105 (Zeiss, Feldbach, Switzerland) color camera was used for image acquisition.

## FTIR
Infrared absorption spectra were measured using a Varian 640-IR spectrometer equipped with diamond attenuated total reflectance (ATR) optics from as prepared hydrogel samples.

The hydrogels were flipped on the crystal to record the spectrum of each layer in the case of layered hydrogels.

## Raman microscopy on gastrointestinal tissue with and without DL-patch
Raman measurements were performed as previously described[47,48]. More specifically, measurements were performed on deparaffinized histological sections on a WITec alpha 300 R confocal Raman microscope, equipped with a UHTS 300 Vis spectrometer and an Andor Newton EMCCD. A linearly polarized 532 nm laser and a laser power of 5 mW were used for excitation. A Zeiss EC Epiplan Neofluar Dic 50x objective (NA 0.8) was used. Spectra were acquired with an integration time of 2 s with a step size of 2 μm and in maps of size ($50 \times 150$ μm$^2$), starting at the outer surface of the gastrointestinal tissue samples. Three maps from different regions were recorded per sample. The preprocessed (cosmic ray removed and background-substracted) spectra were subjected to k-means clustering analysis (Control Four Software, WITec).

## Perfused medium cytotoxicity experiments
Perfused cell medium experiments were performed using a modified procedure inspired from Darnell et al.[41]. More precisely, Normal Human Dermal Fibroblasts (NHDFs, C-12302) (Sigma-Alrdrich, Buchs, Switzerland, C-12300), a non-cancerous human skin fibroblast cell line, was cultured under standard culture conditions at 37 °C with (5%) CO$_2$. Dulbecco's Modified Eagle's Medium – high glucose (DMEM) (#RNBG3787, Sigma, Buchs, Switzerland) supplemented with (10%)

Fetal calf serum (FCS, Sigma-Aldrich, Buchs, Switzerland), (1%) L-Glutamine (Sigma-Aldrich, Buchs, Switzerland) and (1%) Penicillin-Streptomycin (Sigma-Aldrich, Buchs, Switzerland) was used as full growth medium. Respective hydrogel samples (DL-patch, DL-patch mIPN, DL-patch TurnON, DL-patch TurnOFF) were placed inside (15 mL) falcon tubes and brought to swelling equilibrium by being incubated with 5 mL full growth DMEM – high glucose for 2 h. The perfused hydrogel medium was then removed and collected marking the first perfusion iteration. The process was repeated four times. Per 96-plate well (4'000 NHDF) cells were seeded in (100 μL) full growth medium and allowed to attach. At time point 0 h, 100 μL of perfused hydrogel supernatant or full growth medium as control was added respectively (total volume 200 μL) and incubated under standard culture conditions for 24 h. The cell viability was investigated through the LDH dependent CytoTox 96 Non-radioactive Cytotoxicity assay (#G1780, Promega, Dübendorf, Switzerland.) and the ATP dependent CellTiter-Glo® 2.0 Cell Viability Assay (#G9241, Promega, Dübendorf, Switzerland) according to the manufacturer procedure respectively. Untreated cells served as negative control and 0.1% TritonX treated cells served as positive control. At least three independent experiments were carried out per experimental group.

## Intestinal leak sealing model
In order to evaluate the sealant properties of the hydrogels on tissue of the intestine in a most demanding yet highly reproducible scenario allowing quantitative performance evaluation, a cup-based anastomotic leak setup was put in place (see Figure S1). Briefly, a piece of intestine (7 cm) was cut along its length and spread on a piece of flat teflon. The intestine was punctured with a hole simulating an opening at the level of the sutured anastomotic area, using a (4 mm) biopsy punch. Control samples were not equipped with a hole while hydrogels and surgical patches sealed the hole bearing intestine piece, using the previously described procedure. The sealed intestine was then mounted on a bottomless single-use polystyrene cup. The intestine attached cup was subsequently placed on a second polypropylene cup of known mass. The system was then loaded with (7 g) of freshly prepared simulated intestinal fluid or with the same quantity of simulated gastric fluid or phosphate buffer saline. All the samples were placed inside a box with humidified atmosphere and loaded on a platform shaker operating at 40 rpm at 37 °C. The mass of the leaked SIF was then measured at discrete time intervals. At least three independent experiments were carried out per experimental group.

## Digestive leak adhesion evaluation−T-peel
To evaluate adhesion energies under digestive leak conditions a setup was put together to bring patch sealed tissues in contact with digestive effluents in a well-controlled manner. More specifically rectangular hollow molds of dimensions $52 \times 69 \times 8$ mm, depth of 5 mm and inner dimensions $40 \times 60$ mm were designed using TinkerCAD (Autodesk) and 3D printed using a Prusa Mini + (purchased from Prusa research a. s. Czech 2021) and a 1.75 mm Recreus Filaflex Aquamarine flexible polyurethane filament and a 20% infill. The molds were fitted with a rectangular piece of intestine sealed with 3 parallelly placed rectangular patches of dimensions $5 \times 1.5$ cm spaced and applied to fit the dimensions of the hollow part of the mold. The tissue was kept in place by 4 pins and its serosa with the applied sample patches facing upwards attached from the flexible supports. The mold was then filled with 10 mL of the digestive fluid of interest and the system was covered with kitchen foil to avoid drying and maintain 100% humidity. The experiment was initiated by being placed in an oven at 37 °C on top of an orbital shaker (50 rpm).

At the time point of interest, the tissue/patch sample was removed from the oven and was cut and prepared into the shape and form of the samples as described earlier.

## Burst pressure under digestive leak conditions

Burst pressure under conditions of digestive leak was investigated using a 50 ml syringe (Braun, Omnifix) filled with the digestive fluid of interest and mounted onto a syringe pump (KD Scientific, C/N 78-9410). These were connected with a custom-made chamber apparatus created following design criteria elaborated by Mooney et al[22]. and of inner diameter of 31 mm. The chamber delimited a fillable volume of 15 mL and was connected to a pressure sensor (Honeywell, ABP-DANV060PGSA3). Experiments were performed at flow rates of 2 mL/min. Data were then collected onto a csv file.

## Ultrasound suture breach evaluation model

Tubular pieces of freshly thawed small porcine intestine were cut into 7 cm long pieces. With the help of the rear end of a plastic scalpel and a 4 mm biopsy punch the tissues were perforated at the middle point of the tissue, simulated this way a breach at the level of sutures. The samples were then sealed with a double-layered hydrogel of choice (as earlier described) and one of their extremities was closed to aid with fluid loading using a plastic zip tie. The tissue-patch system was then lifted and secured on tweezers using a holder. At this stage it was made possible to load the tissue with at least 15 mL of digestive fluid (SGF or SIF) and subsequently seal the tissue with a second zip tie. The system was then placed in transparent foil and incubated for discrete time intervals at 37 °C on an orbital shaker of 20 rpm. At the desired time point samples were removed and placed in a plastic box of 6 cm depth which was filled with ultrasound gel after positioning sealant down of the sample. The gel filled recipient was then controlled for large air bubbles, which were removed by the addition of water or the use of a 200 μL pipette. Subsequently the system was sealed using transparent foil and a Clarius 7 HD (purchased from Meditron SA. Morges Switzerland) in combination with an ipad (Apple California 2019) probe was used for ultrasound monitoring. At least three independent experiments were carried out per experimental group.

## In vivo study in a piglet model

An in vivo study certified by the Commission of Work with Experimental Animals at the Medical Faculty of Pilsen, Charles University (project ID: MSMT-15629/2020-4), and under control of the Ministry of Agriculture of the Czech Republic has been performed. All procedures were performed in compliance with the law of the Czech Republic and in line with the legislation of the European Union. A healthy male Prestice black-pied pig was pre-medicated using a mixture of ketamine (Narkamon 100 mg/mL, BioVeta a.s., Ivanovice na Hané, Czech Republic) and azaperone (Stresnil 40 mg/mL, Elanco AH, Prague, Czech Republic) administered intramuscularly. General anesthesia was maintained by continuous intravenous administration of propofol (Propofol 2% MCT/LCT Fresenius Medical Care a.s.). Nalbuphin (Nalbuphin, Torrex Chiesi CZ s.r.o., Czech Republic) which was used intravenously for analgesia assurance. We entered the abdominal cavity via a midline laparotomy. A 1 cm large incision, approximately, was made using electrocoagulation on the small intestine roughly 50 cm aborally from the duodeno-jejunal transition. A second 1 cm large defect was made at the level of the front wall of the stomach in the same fashion. The defects were then sealed with a rectangular layered patch in the case of the stomach and circular layered patches for the small intestine region (all containing equally spaced TurnOFF sensing elements of 5 μL each). A single circular hydrogel patch sealant was attached to a non-defective small intestine region in the opposite side of the abdominal cavity to prevent potential contamination with gastrointestinal contents. The abdominal cavity was temporarily closed using PDS monofilament 2/0 suture (polydioxanone, Ethicon Inc., New Jersey, USA). Immediately after reconstruction of the abdominal wall, sonography was performed using a Clarius 7 HD (purchased from Meditron SA. Morges Switzerland) to locate the hydrogels in the abdominal cavity and to rule out false positivity of the

diagnostic compound. The animal was monitored for 120 min, at which point a second sonography was performed to identify the hydrogel sealants as well as their changes in terms of echogenicity of the sensing element and location. At this point the sutured laparotomy was re-opened and the abdominal cavity was explored. Tissue samples of the intestinal wall and stomach wall were collected (1. Normal stomach wall, 2. Stomach wall from the site of application of the hydrogel, 3. Normal intestinal wall, 4. Intestinal wall from the site of application of the hydrogel) and fixed in buffered formalin for histological evaluation. The animal was then immediately terminated after sample collection using a cardioplegic solution. Photodocumentation was taken throughout the experiment.

## Statistics and reproducibility

At least three independent experiments were performed per condition. Specific details are indicated in the figure captions. Experiments were repeated over time periods of several months and performed by different operators in order to ensure repeatability. Standards in the field were followed and no statistical method was used to predetermine sample size. No data were excluded from the analyses.

## Reporting summary

Further information on research design is available in the Nature Research Reporting Summary linked to this article.

## Data availability

All the data supporting the findings of this study are available within the main text of this article and its Supplementary Information, or from the corresponding authors upon request. Source data are provided with this paper.

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

## Acknowledgements

We acknowledge financial support from the Swiss National Science Foundation (Eccellenza grant. 181290, I.K.H.), the SwissLife Foundation (I.K.H.), the Vontobel Foundation (I.K.H.), the Claude et Giuliana Foundation (I.K.H.), the Hans Gröber Foundation (I.K.H.), the Evi-Diethelm-Winteler-Foundation (I.K.H.) and the Dornonville-DeLaCour Foundation (I.K.H.). M.G.S. is supported by the US National Institutes of Health (R01EB018975, M.G.S.) and the Jacobs Institute for Molecular Engineering in Medicine (M.G.S.). M.P.A. is supported by the Agency for Science, Technology and Research of Singapore. A.H.C.A is supported by an ETH Pioneer Fellowship. We thank Apolline Anthis for her contributions towards the development of the layer-by-layer incorporation of sensing and therapeutic elements in model patches and Anna-Katharina Zehnder for her contributions towards evaluating burst pressure under digestive conditions. We thank Michel Calame for access to the Raman microscope, the Empa Biointerfaces lab for access to the bacteria lab and the Empa Electron Microscopy Center for access to the SEM. The schematics and pictograms in the figures were created using BioRender.com.

## Author contributions

A.H.C.A. contributed to study design, developed, and identified all the polymer and functional element formulations, performed experiments, analysed data and drafted the manuscript. M.P.A. produced GVs, provided guidance in working with GVs. A.L.N. performed cytocompatibility analysis. E.T. performed electron microscopy. J.R. performed animal surgeries. T.R. provided crucial input for ultrasound imaging. F.H.L Starsich synthesized the zinc oxide nanoparticles. B.W. supervised and performed the tensile tests together with A.H.C.A. V.L. supervised the animal work. A.A.S. provided clinical input and contributed to study design. M.S. supervised the GV production and provided guidance in working with GVs. I.K.H. conceived, designed and supervised the study, performed the Raman spectromicroscopy and co-wrote the manuscript. All authors contributed to discussions and edited the manuscript.

## Competing interests

A.H.C.A. and I.K.H. declare that patent applications have been filed covering all parts of the adhesion and sensing technology reported in this publication. The patents have been filed by ETH Zurich and Empa (Alexandre H.C. Anthis, Martin T. Matter and Inge K. Herrmann, PCT/EP2022/051137, patent field and Alexandre H.C. Anthis, Benjamin Suter and Inge K. Herrmann, PCT/EP2022/051141, patent filed). All other authors report no conflict of interest.
