## [Peer Review File · Nature Communications]

Modular stimuli-responsive hydrogel sealants for early gastrointestinal leak detection and containmentReviewer #1 (Remarks to the Author):

Reviewer's Comments: The manuscript entitled, 'Modular stimuli-responsive hydrogel sealants for early gastrointestinal leak detection and containment', reports on modular stimuli-responsive surgical hydrogel sealants leveraging tissue penetrating polymer networks and trigger-responsive echogenic entities to enable point-of-need monitoring and early anastomotic leak detection using a hand-held ultrasound transducer and a smartphone. Moreover, reliable detection of the breaching of sutures, in as little as 3 hours in intestinal leak scenarios and 15 minutes in gastric leak conditions, is demonstrated. This manuscript can't be considered for publication in its present form. The following points need to address before consideration.

Point 1. All the abbreviations used in the article must be expanded in full and defined form in their first appearance and subsequently only abbreviated form would be used for more readability of the audience.

Point 2. In hydrogel preparation and assembly subsection authors stated that 108 μL of mBAA crosslinker stock solution was used- why it is so specific?

Point 3. The manuscript needs to recheck for making it more lucid language. For example, 'From this solution were prepared 20% vol gas vesicle (Ana or Halo) dispersions which served as TurnOFF sensing elements.'

Point 4. The characteristics like size, shape, aspect ratio, etc. of ZnO nanoparticles are necessary to mention in the manuscript, as they can influence the performance. Additionally, is there any synergistic effect of such nanoparticles with the sensing element?

Point 5. Authors claimed antimicrobial activity of ZnO nanoparticles just by testing them against E. coli bacterial strain, which is not justified! So, they have to show the activities against different bacteria, fungi, and consortium of alga to authenticate their claim.

Point 6. The studied patch sealant is necessary to administrate to the host through surgery, which is not only painful but may also cause secondary infection! Can it be administrated through injection? What would be the associated issues? Please discuss.

Point 7. Is this patch sealed for minor surgery without using suture or stapler? In such case what would be the maximum dimension of the surgical operation?

Point 8. Is this sealing patch biodegradable? If so, are the byproducts, particularly leak monitoring elements not-toxic? If not, then second surgery is necessary for its removal, which may result delaying of healing process! Please confirm.

Point 9. Authors claimed that 'The unique leak-detection and tissue monitoring capabilities incorporated within the sealant patch operate based on matrix engineered pH and/or enzyme-responsive triggerable sensing elements and can be read out by non-invasive point-of-need ultrasound imaging.'-Mechanism need to be elaborated in support of the statement.

Point 10. What are the required characteristics of the used mutually interpenetrating network to be used for such purposes? Need to be mentioned.

Reviewer #2 (Remarks to the Author):

The authors report a stimuli-responsive hydrogel sealant for early detection of gastrointestinal leak after surgery. Overall, the characterization on stimuli-responsive properties of the sealant was well performed. However, its adhesive properties as a "sealant" requires more in-depth study.

The development of sealants for the gastrointestinal tract is reported to be the most difficult compared to other tissues due to the dynamic environment, as noted by the authors. The adhesion required for gastrointestinal tissue, in particular, in the presence of biofluids can be different from that on normal tissues, and this is the reason why there are few reports on sealants for gastrointestinal tissues compared to other tissues. For example, many conventional sealants lose their adhesion in the presence of gastrointestinal fluids. Therefore, adhesion tests by T-peel setup and tensile characterization are insufficient to describe the adhesive properties of the sealant the authors developed. Adhesion should be characterized over long periods of time under

the dynamic conditions (i.e., in the presence of flowing biofluid). In addition, the burst pressure due to fluid leakage must also be considered during this characterization.

Meanwhile, long-term in vivo degradation/clearance and toxicity to nearby tissues should also be characterized unless the sealant is surgically removed.

Reviewer #3 (Remarks to the Author):

Anthis et al have proposed a very noteworthy novel method of seemingly both potentially mitigating against and providing early warning of anastomotic leak. If their porcine model can be made applicable to humans this translational research will be of immense significance as anastomotic leaks are major drivers of increased morbidity, mortality, and cost. I am unaware of any similar work in this realm. I do believe that their work supports their conclusions and claims although there is much needed work to be done if the model is to be successful in humans. For now, I think that this work could certainly be published to allow people to begin to become familiar with this technique. The methodology does appear sound and the work seems to meet the expected standards with sufficient detail provided. There are some minor issues such as the fact that the authors cite a 10% rate of anastomotic leak. They should be more cognizant that although the 10% is an average there are parts of the intestinal tract such as the esophagus and the rectum in which these leak rates are much higher. Secondly, the effect of both digestive enzymes in the esophagus and bacterial flora in the colon and rectum might challenge the utility, efficacy, and reliability of this model. The authors would need to test their modular stimuli-responsive hydrogel sealants in a variety of gastrointestinal milieu. An additional question is would the hydrogel potentially prevent anastomotic leaks and if so would the long term sequelae of preventing a leak in a faulty anastomosis result in an anastomotic stricture? This is another detail that would subsequently need to be analyzed. However, for now this study is indeed very intriguing, quite novel, and I would be of significant interest to readers.

Letter of Reply

We thank the reviewers for their careful and positive assessment of our manuscript. Please find a detailed reply to the respective comments and concerns below. Changes to the manuscript text are **highlighted**.

Reviewer #1 (Remarks to the Author):

Reviewer's Comments: The manuscript entitled, 'Modular stimuli-responsive hydrogel sealants for early gastrointestinal leak detection and containment', reports on modular stimuli-responsive surgical hydrogel sealants leveraging tissue penetrating polymer networks and trigger-responsive echogenic entities to enable point-of-need monitoring and early anastomotic leak detection using a hand-held ultrasound transducer and a smartphone. Moreover, reliable detection of the breaching of sutures, in as little as 3 hours in intestinal leak scenarios and 15 minutes in gastric leak conditions, is demonstrated. This manuscript can't be considered for publication in its present form. The following points need to address before consideration.

Point 1. All the abbreviations used in the article must be expanded in full and defined form in their first appearance and subsequently only abbreviated form would be used for more readability of the audience.

We thank the reviewer for raising this point and have addressed the abbreviation issue as suggested.

Point 2. In hydrogel preparation and assembly subsection authors stated that 108 μL of mBAA crosslinker stock solution was used- why it is so specific?

We thank the reviewer for raising this point. The text has now been amended for reasons of clarity. The amount of 108 μL of mBAA crosslinker stock solution reflects the creation of a hydrogel with a monomer to crosslinker ratio of 2000. This is a hydrogel basis that based on our previous experience and adjustments over published literature (e.g., Li, J. et al. *Science* 357, 378–381 (2017)) yields polyacrylamide hydrogels with desirable properties. Changes to the manuscript can be found on page 22 and the paragraph "Hydrogel preparation and assembly"

Point 3. The manuscript needs to recheck for making it more lucid language. For example, 'From this solution were prepared 20% vol gas vesicle (Ana or Halo) dispersions which served as TurnOFF sensing elements.'

The entire manuscript text has now been checked and amended for clearer language where appropriate.

Point 4. The characteristics like size, shape, aspect ratio, etc. of ZnO nanoparticles are necessary to mention in the manuscript, as they can influence the performance. Additionally, is there any synergistic effect of such nanoparticles with the sensing element?

We thank the reviewer for raising this point. Figure S6 has now been added to the manuscript including particle characteristics namely size (including geometric standard deviation σ_g 1.4), specific surface area and crystallinity.

Properties of ZnO NPs

BET	30 m ² /g
XRD	Crystallinity: wurzite d _{XRD} : 21nm
TEM	d _{TEM} : 24 nm (σ _g : 1.4)

Figure S6: ZnO nanoparticles characterization used as a model inorganic antimicrobial. Transmission electron microscopy (TEM) images depicting ZnO nanoparticles (top left, scale bar: 20 nm) and high resolution TEM images of selected ZnO nanoparticle (top right, scale bar 5 nm). Table summarizing the surface area, crystal structure and size of used nanoparticles.

Regarding potential synergistic effects between ZnO nanoparticles and sensing elements, while a very interesting point underlined by this comment, in this article we have tried to maintain discretized therapeutic elements and sensing elements as only the latter are central to the idea of early detection of leaks (prior to complications). However, further investigations could give rise to materials that not only release therapeutic elements upon leak occurrence but also upon demand and patient rehabilitation schedule (Shapiro et al¹). This potential future prospect has now been added to the discussion and can be found on page 20:

“Additionally, the demonstrated and illustrative combination of sensing moieties with therapeutic elements such as antibacterials (ZnO, or gentamicin) gave rise to adhesives that could both detect but also potentially prevent infection in a stimuli responsive manner via the release of therapeutics. Continued investigations into synergies of those elements could give rise suture supports that release therapeutic payloads following an external trigger¹.”

1. Bar-Zion, A. et al. Acoustically triggered mechanotherapy using genetically encoded gas vesicles. *Nat. Nanotechnol.* **16**, 1403–1412 (2021).

Point 5. Authors claimed antimicrobial activity of ZnO nanoparticles just by testing them against E. coli bacterial strain, which is not justified! So, they have to show the activities against different bacteria, fungi, and consortium of alga to authenticate their claim.

We agree with the reviewer that testing ZnO against E. coli is not sufficient to claim overall antimicrobial activity. However, the antimicrobial effects of ZnO nanoparticles have been widely investigated already by many against a large selection of microbes^{1,2} and also algae³. For the exact same batch of flame-made- ZnO nanoparticles, we have measured antimicrobial activity against E. coli, S. aureus (MRSA) and C. albicans (see Figure below), to confirm that they exhibit the properties found by others for ZnO.

To avoid a misunderstanding, we adapted the text in order to make it clear that ZnO is only used as an example to illustrate that, these patches may also exhibit additional functions (page 6):

“...properties imparted to the system following the illustrative embedding of...”

and page 12:

"The same properties also allow the introduction of therapeutics (such as antimicrobial^[35,36] ZnO nanoparticles, see Figure S6 for characterization data),"

1. Pasquet, J. *et al.* The contribution of zinc ions to the antimicrobial activity of zinc oxide. *Colloids and Surfaces A: Physicochemical and Engineering Aspects* **457**, 263–274 (2014).
2. da Silva, B. L. *et al.* Relationship between structure and antimicrobial activity of zinc oxide nanoparticles: An overview. *International journal of nanomedicine* **14**, 9395 (2019).
3. Suman, T. Y., Rajasree, S. R. & Kirubakaran, R. Evaluation of zinc oxide nanoparticles toxicity on marine algae *Chlorella vulgaris* through flow cytometric, cytotoxicity and oxidative stress analysis. *Ecotoxicology and environmental safety* **113**, 23–30 (2015).

Point 6. The studied patch sealant is necessary to administrate to the host through surgery, which is not only painful but may also cause secondary infection! Can it be administrated through injection? What would be the associated issues? Please discuss.

We thank the reviewer for raising these important points. Indeed, the idea at present is that the hydrogel patch is applied intraoperatively during the routinely performed open abdomen anastomotic procedure. Thus, the application of the hydrogel patch will not require any additional surgery but could be applied intraoperatively as a suture support material. However, in the future, an application during laparoscopic (or robotic) procedures is indeed appealing and is therefore the focus of future research. In order to reflect these points more explicitly in the manuscript, we have made the following addition (page 5):

“The envisaged application of the leak-detecting hydrogel sealant patch, during a resection and anastomosis surgery, enabling...”

Point 7. Is this patch sealed for minor surgery without using suture or stapler? In such case what would be the maximum dimension of the surgical operation?

Based on discussions with expert surgeons, we consider it at this stage unlikely that such a patch could replace sutures or staples in the near future. Importantly, sutures and staples can also act as stable in-tissue anchors avoiding complete tissue disconnection and subsequent spillage of intestinal content into the abdomen. However, reinforcement of sutured or stapled tissue sites with a support material, such as the developed patch, is highly desirable to prevent leaking from small holes formed during the suturing or stapling in patients with impaired healing.

We have added the following discussion on page 18:

“These latter also underscore and foreshadow the important role of sutures during events of deterioration, as in the inevitable case of a leak, they allow the limitation of leaked fluid volumes. Thus, while the development of technologies capable of offering sutureless surgeries,

abdominal cavity operations shall be seen as in need for suture support rather than suture replacement.”

Point 8. Is this sealing patch biodegradable? If so, are the byproducts, particularly leak monitoring elements not-toxic? If not, then second surgery is necessary for its removal, which may result delaying of healing process! Please confirm.

This is indeed central point of discussion. The current patch is designed in such a way that it can contain leaks when they occur, and thus is non-digestible. It is synthetically demanding to produce a material, that can at first resist gastrointestinal fluids and contain a leak without being degraded/digested, and then later biodegrades when no longer needed. The sensing elements are not considered to be high risk; in one case it is carbonates (which are inherently biocompatible), and the GVs have successfully been employed in vivo without any detectable complications (Shapiro et al¹).

We feel that it is important to point out that patients with leaks and poorly healing anastomoses are at imminent danger of sepsis and death, and thus being able to put a material to contain the leak and seal the defect is imperative for their survival. However, we agree, that long-term fate and biodegradation are key targets for future research. We have added the following discussion to the manuscript page 20:

“...As such and while further animal studies and material optimization (fate and biodegradation) are envisaged,..”

1. Shapiro, M. G. et al. Biogenic gas nanostructures as ultrasonic molecular reporters. *Nature Nanotech* **9**, 311–316 (2014).

Point 9. Authors claimed that ‘The unique leak-detection and tissue monitoring capabilities incorporated within the sealant patch operate based on matrix engineered pH and/or enzyme-responsive triggerable sensing elements and can be read out by non-invasive point-of-need ultrasound imaging.’- Mechanism need to be elaborated in support of the statement.

Please be referred to SI Figure S7 showing the enzymatic digestion of the proteinaceous capsules of the GVs, and Figure 3 showing the pH-dependent dissolution and corresponding CO₂ bubble formation for the bicarbonate elements.

Point 10. What are the required characteristics of the used mutually interpenetrating network to be used for such purposes? Need to be mentioned.

This is indeed a topic of central importance, and we thank the reviewer for raising the point. A suitable mIPN needs to be unchanging in mechanical properties during conditions of leaks, all while not harming the tissue upon contact. We have added the following paragraph to the manuscript text on page 17:

“Taken all the above together, it is also important to observe some of the prevailing qualities of an mIPN designed for the purposes of sealing tissues under digestive leak conditions. These can be discretized as (i) being mechanically stable under harsh digestive conditions, (ii) having little to no interaction with the functioning of the patch or its components, (iii) having a composition that is not disruptive to tissue and the sealing interface.”

Reviewer #2 (Remarks to the Author):

The authors report a stimuli-responsive hydrogel sealant for early detection of gastrointestinal leak after surgery. Overall, the characterization on stimuli-responsive properties of the sealant was well performed. However, its adhesive properties as a “sealant” requires more in-depth study.

The development of sealants for the gastrointestinal tract is reported to be the most difficult compared to other tissues due to the dynamic environment, as noted by the authors. The adhesion required for gastrointestinal tissue, in particular, in the presence of biofluids can be different from that on normal tissues, and this is the reason why there are few reports on sealants for gastrointestinal tissues compared to other tissues. For example, many conventional sealants lose their adhesion in the presence of gastrointestinal fluids. Therefore, adhesion tests by T-peel setup and tensile characterization are insufficient to describe the adhesive properties of the sealant the authors developed. Adhesion should be characterized over long periods of time under the dynamic conditions (i.e., in the presence of flowing biofluid). In addition, the burst pressure due to fluid leakage must also be considered during this characterization.

We thank the reviewer for raising these important issues and we fully agree. We have investigated the sealant properties of the patch in a, according to our surgical collaborators, most demanding model system (see SI Figure S1). In this experiment, the patch was attached to intestinal tissue and used to seal a 4 mm open defect, while being constantly exposed to intestinal fluid and movement. In this highly demanding experiment, sealing performance of the patch was significantly better compared to Tachosil, and indistinguishable from defect-free control intestine for periods of at least 24 hours (ex vivo). Tensile characteristics in presence of intestinal and gastric fluid can be found in Figure 3c.

Based on the reviewer's request, we have now complemented these data with additional burst pressure data and T-peel experiments after intestinal fluid exposure (in an extreme scenario of up to 8 hrs direct digestive fluid contact) and have added the following paragraph and data to the manuscript page 15:

“Further light into the behaviour of the sealant patches under digestive conditions was shed using burst pressure and additional T-peeling experiments performed under digestive conditions (see Figure S1c,d). Adhesion under those conditions, including shaking and immersive digestive fluid contact was observed to retain at least 50% of its initial adhesion after 8h of incubation under enzymatically active intestinal fluid contact, while in the case of gastric effluents (pH of 2.0), values remained close to levels upon application. Lastly, burst pressure was also investigated under conditions of digestive fluid contact, showcasing similar maximal burst pressures for both SGF and SIF conditions, overall greatly surpassing the native maximal intestinal burst pressure.”

Meanwhile, long-term in vivo degradation/clearance and toxicity to nearby tissues should also be characterized unless the sealant is surgically removed.

We agree with the reviewer that this is an important point. The current patch is designed in such a way that it can contain leaks when they occur and home sensing elements, and thus is non-digestible. It is synthetically demanding to produce a material, that can at first resist gastrointestinal fluids and contain a leak without being degraded/digested, and then later biodegrades when no longer needed.

We feel that it is important to point out that patients with leaks and poorly healing anastomoses are at imminent danger of sepsis and death, and thus being able to put a material to contain the leak and seal the defect is imperative for their survival. However, we agree, that long-term fate and biodegradation are key targets for future research, as now mentioned also in the manuscript on page 20.

“...optimization (fate and biodegradation) are envisaged...”

Reviewer #3 (Remarks to the Author):

Anthis et al have proposed a very noteworthy novel method of seemingly both potentially mitigating against and providing early warning of anastomotic leak. If their porcine model can be made applicable to humans this translational research will be of immense significance as anastomotic leaks are major drivers of increased morbidity, mortality, and cost. I am unaware of any similar work in this realm. I do believe that their work supports their conclusions and claims although there is much needed work to be done if the model is to be successful in humans. For now, I think that this work could certainly be published to allow people to begin to become familiar with this technique. The methodology does appear sound and the work seems to meet the expected standards with sufficient detail provided. There are some minor issues such as the fact that the authors cite a 10% rate of anastomotic leak.

1. They should be more cognizant that although the 10% is an average there are parts of the intestinal tract such as the esophagus and the rectum in which these leak rates are much higher.

We thank the reviewer for the kind evaluation of our manuscript and for raising this point. The 10% mentioned in the abstract refers to an average. We have discussed potentially higher leak incidence rates and causes in the introduction.

2. Secondly, the effect of both digestive enzymes in the esophagus and bacterial flora in the colon and rectum might challenge the utility, efficacy, and reliability of this model. The authors would need to test their modular stimuli-responsive hydrogel sealants in a variety of gastrointestinal milieu.

We have added a statement regarding the importance of different gastrointestinal milieus and patient variability on page 15 of the manuscript.

“While these results are not fully representative of the even greater variety of gastrointestinal milieus present in different patients with different conditions, it is notable that these come in stark contrast to Tachosil sealed holes, which leak within minutes in the settings^[13] explored and offer no diagnostic feature.”

3. An additional question is would the hydrogel potentially prevent anastomotic leaks and if so would the long term sequelae of preventing a leak in a faulty anastomosis result in an anastomotic stricture? This is another detail that would subsequently need to be analyzed.

We agree that this is a point, that needs to be carefully investigated in long-term with in vivo experiments.

However, for now this study is indeed very intriguing, quite novel, and I would be of significant interest to readers.

We thank the reviewers for their feedback and support.

Reviewer #1 (Remarks to the Author):

Based on the responses of the authors to the comments of the reviewers, I recommend the manuscript for consideration.

Reviewer #2 (Remarks to the Author):

The revised manuscript is successful in clearly answering the reviewer's comments. As a result, this manuscript is recommended to publication with no need to further review.

Reviewer #3 (Remarks to the Author):

I read the revision of the manuscript NCOMMS-22-07392-A. The authors appear to have made the appropriate edits in the manuscript. Accordingly, I recommend proceeding with publication.